# NKX2-5 regulates human cardiomyogenesis via a HEY2 dependent transcriptional network

David J. Anderson[1], David I. Kaplan[2], Katrina M. Bell[1], Katerina Koutsis[1], John M. Haynes[3], Richard J. Mills[4], Dean G. Phelan[1], Elizabeth L. Qian[1], Ana Rita Leitoguinho[1], Deevina Arasaratnam[1], Tanya Labonne[1], Elizabeth S. Ng[1], Richard P. Davis[5], Simona Casini[5], Robert Passier[5], James E. Hudson[4], Enzo R. Porrello[4], Mauro W. Costa[6], Arash Rafii[7,8], Clare L. Curl[9], Lea M. Delbridge[9], Richard P. Harvey[10,11], Alicia Oshlack [1], Michael M. Cheung[1,12], Christine L. Mummery[5], Stephen Petrou[2], Andrew G. Elefanty[1,12,13], Edouard G. Stanley[1,12,13] & David A. Elliott [1,14,15]

Congenital heart defects can be caused by mutations in genes that guide cardiac lineage formation. Here, we show deletion of NKX2-5, a critical component of the cardiac gene regulatory network, in human embryonic stem cells (hESCs), results in impaired cardiomyogenesis, failure to activate VCAM1 and to downregulate the progenitor marker PDGFRα. Furthermore, NKX2-5 null cardiomyocytes have abnormal physiology, with asynchronous contractions and altered action potentials. Molecular profiling and genetic rescue experiments demonstrate that the bHLH protein HEY2 is a key mediator of NKX2-5 function during human cardiomyogenesis. These findings identify HEY2 as a novel component of the NKX2-5 cardiac transcriptional network, providing tangible evidence that hESC models can decipher the complex pathways that regulate early stage human heart development. These data provide a human context for the evaluation of pathogenic mutations in congenital heart disease.

[1] Murdoch Childrens Research Institute, Royal Children's Hospital, Flemington Road, Parkville, VIC 3052, Australia. [2] The Florey Institute of Neuroscience and Mental Health; Centre for Neuroscience, University of Melbourne, Parkville, VIC 3052, Australia. [3] Monash Institute of Pharmaceutical Science, Monash University, 381 Royal Parade Parkville, Victoria 3052, Australia. [4] School of Biomedical Sciences, University of Queensland, Brisbane, QLD 4072, Australia. [5] Department of Anatomy and Embryology, Leiden University Medical Center, Albinusdreef 2, 2333 ZA Leiden, The Netherlands. [6] The Jackson Laboratory, Bar Harbor, ME 04609, USA. [7] Stem Cell and Microenvironment Laboratory, Weill Cornell Medical College in Qatar Qatar Foundation, Doha, Qatar. [8] Department of Genetic Medicine, Weill Cornell Medical College, New York, NY, USA. [9] Department of Physiology, University of Melbourne, Parkville, VIC 3052, Australia. [10] Victor Chang Cardiac Research Institute, Darlinghurst, NSW 2052, Australia. [11] St. Vincent's Clinical School and School of Biotechnology and Biomolecular Sciences, University of New South Wales, Kensington 2052, Australia. [12] Department of Pediatrics, The Royal Children's Hospital University of Melbourne Parkville VIC 3052 Australia. [13] Department of Anatomy and Developmental Biology, Faculty of Medicine, Nursing and Health Sciences, Monash University, Clayton, VIC 3800, Australia. [14] Australian Regenerative Medicine Institute, Monash University, Clayton, VIC 3800, Australia. [15] School of Biosciences, University of Melbourne, Parkville, VIC 3052, Australia. Correspondence and requests for materials should be addressed to D.A.E. (email: david.elliott@mcri.edu.au)

Perturbations of the gene regulatory networks (GRNs) that guide lineage formation during human cardiogenesis cause congenital heart defects (CHDs)[1]. The core unit controlling heart development consists of highly conserved transcription factors in a GRN known as the cardiac kernel[2]. Mutations in cardiac kernel members, such as GATA4, NKX2-5, and TBX5, underlie a range of CHDs[3–5]. NKX2-5 encodes an NK-2 class homeodomain protein that is a critical component of the cardiac kernel in all vertebrates studied[6]. In humans, dominant mutations in NKX2-5 cause a range of CHDs, mainly atrioventricular block and atrial septal defects, with a spectrum of other structural conditions such as ventricular septal defect and tetralogy of Fallot at lower frequency[6]. In mice, deletion of Nkx2-5 blocks cardiac looping due to impaired progenitor specification in the second heart field[7] and impairs ventricular chamber morphogenesis resulting in embryonic lethality[7–9]. In addition, introduction of dominant negative Nkx2-5 variants in the mouse causes similar phenotypes to those observed in patients with NKX2-5 mutations, such as AV block and atrial septal anomalies[10,11]. However, the pleiotropic cardiac pathologies associated with NKX2-5 mutations, in both mouse and human, suggest that expression of the NKX2-5 target gene set is further modulated by interaction with available co-factors at a given genomic location[12–14].

To study the role of NKX2-5 in the cardiac GRN and human cardiac development, we investigate cardiac differentiation in vitro using a suite of genetically modified hESCs. We show that NKX2-5 is required to complete cardiomyogenesis and that hESC-derived cardiomyocytes (hESC-CMs) lacking NKX2-5 have compromised expression of cardiac differentiation markers, electrophysiology and contractile function. Gene expression profiling and ChIP-seq identifies HEY2, a NOTCH-dependent bHLH class transcription factor[15], as a potential downstream mediator of NKX2-5. Furthermore, genetic rescue experiments show that HEY2 restores, in part, the cardiac muscle genetic program in NKX2-5 null cardiomyocytes.

## Results

### NKX2-5 regulates cardiac progenitor cell differentiation.

To investigate NKX2-5 function we targeted the wildtype NKX2-5 allele of the heterozygous HES3 NKX2-5$^{eGFP/w}$ line[16]. The resultant null NKX2-5$^{eGFP/eGFP}$ hESC line (denoted NKX2-5$^{-/-}$) was karyotypically normal, expressed pluripotency markers and differentiated into all three germ layers (Fig. 1a, Supplementary Fig. 1a–e). As expected, cardiac cells derived from NKX2-5$^{-/-}$ hESCs expressed GFP (Fig. 1b), but did not produce NKX2-5 protein whereas NKX2-5 levels were comparable between NKX2-5$^{eGFP/w}$ and wildtype cells (Supplementary Fig. 1f). When differentiated to the cardiac lineage as monolayers, NKX2-5$^{-/-}$ hESCs formed GFP$^+$ cells with similar kinetics to the parental NKX2-5$^{eGFP/w}$ line and, by day 14 of differentiation, both cultures contained similar proportions of GFP$^+$ and ACTN2$^+$ cells (Fig. 1b, c and see Supplementary Fig. 1g, h for representative FACS plots). However, the percentage of GFP$^+$ cells was consistently lower in NKX2-5$^{-/-}$ cultures at early time points (Fig. 1c), possibly resulting from disruption of an NKX2-5 autoregulation loop[17]. When differentiated as embryoid bodies, the onset of spontaneous contractility of NKX2-5$^{-/-}$ cultures was similarly delayed but not abrogated (Supplementary Fig. 1i), indicating that human NKX2-5 is not essential for cardiomyocyte contractility, consistent with murine studies[8]. Furthermore, differentiated NKX2-5$^{-/-}$ cultures expressed known cardiomyogenic markers, including TBX5, GATA4, and MYH6, at comparable levels to NKX2-5$^{eGFP/w}$ cultures (Fig. 1d). Despite these delays in the onset of contractility and reduced proportion of early GFP expressing cells, superficially, cardiac differentiation of NKX2-5$^{-/-}$ cultures appeared normal.

Flow cytometry analysis revealed both NKX2-5$^{eGFP/w}$ and NKX2-5 null GFP positive populations were heterogenous, with low GFP expressing cells representing cardiac precursors and non-myocytes (Supplementary Fig. 1g)[18–22]. In addition, NKX2-5$^{-/-}$-derived GFP$^+$ cells retained expression of PDGFRα, a marker of cardiac progenitor cells required for heart tube formation[23], normally downregulated during heart development[7,24]. GFP$^+$ cells from differentiating cultures of both NKX2-5$^{eGFP/w}$ and NKX2-5$^{-/-}$ cells expressed PDGFRα at day 14 (Supplementary Fig. 1j), but after extended culture to day 42, few NKX2-5$^{eGFP/w}$ GFP + cells expressed PDGFRα (9.5 ± 2.6%, n = 5) whereas expression was maintained in NKX2-5$^{-/-}$ GFP + cells (81.4 ± 3.0%, n = 5) (Fig. 1e, f). This is consistent with the enduring and spatially expanded domain of Pdgfrα expression observed in Nkx2-5 knockout mice, resulting from a failure to repress a number of cardiac progenitor-expressed genes[7]. Thus, perdurance of PDGFRα expression suggests incomplete differentiation of NKX2-5 null cardiac cells. These data were complemented by a reduced percentage of VCAM1$^+$ cardiomyocytes in differentiating NKX2-5$^{-/-}$ cultures (Fig. 1g, h). Further, this cell surface marker phenotype is recapitulated in H9 hESCs in which NKX2-5 has been deleted (NKX2-5$^{eGFP/del}$; Supplementary Fig. 1k–m). Given that VCAM1 marks myocardial commitment[18], this data also suggested a block in cardiomyogenesis in the absence of NKX2-5. In summary, cardiac differentiation of NKX2-5$^{-/-}$ hESCs yielded contractile cardiomyocytes, but reciprocally altered expression of VCAM1 and PDGFRα implies perturbed differentiation.

### Impaired function of NKX2-5$^{-/-}$ cardiomyocytes.

NKX2-5 null monolayer cardiomyocyte cultures displayed abnormal patterns of contraction (Supplementary Movie 1). We correlated calcium oscillations during contraction between adjacent areas in sheets of beating cardiomyocytes, and demonstrated that NKX2-5$^{eGFP/w}$ cardiac sheets showed greater synchronicity of contraction (correlation co-efficient, $R^2$, 0.69 ± 0.10, n = 5) than NKX2-5 null cardiomyocyte monolayers ($R^2$, 0.23 ± 0.09, n = 5) (Fig. 2a, b). The maximal amplitude of calcium flux was also much higher in NKX2-5$^{eGFP/w}$ cultures (Fig. 2c), suggesting that calcium handling of NKX2-5 null cardiomyocytes was either defective or had not reached an equivalent level of maturation.

Multi-electrode array (MEA) analysis showed that NKX2-5$^{eGFP/w}$ and NKX2-5$^{-/-}$ cardiac aggregates had a similar basal rate of contraction (Fig. 2d, e). However, NKX2-5$^{-/-}$ cardiac aggregates exhibited a prolonged field potential duration at both early (112 ± 7 ms in NKX2-5$^{ew}$ cells (283 ± 34 μN/w vs. 257 ± 12 ms in NKX2-5$^{-/-}$, n = 21, p < 0.0001; Fig. 2d,f) and late stages of differentiation (Supplementary Fig. 2a). Similarly, whole-cell patch clamp analysis of spontaneously contracting single cells demonstrated a similar rate of contraction between individual NKX2-5$^{eGFP/w}$ and NKX2-5$^{-/-}$ cardiomyocytes (Supplementary Fig. 2b, c), but prolonged action potential durations in individual NKX2-5$^{-/-}$ cardiomyocytes at the same contraction rate (APD90 229 ± 21 ms in NKX2-5$^{eGFP/w}$ vs. 429 ± 34 ms in NKX2-5$^{-/-}$ n = 8, p < 0.0001; Supplementary Fig. 2b, d). The initial upstroke velocity of NKX2-5$^{-/-}$ cardiomyocytes was also slower than that of NKX2-5$^{eGFP/w}$ cardiomyocytes (4.1 ± 0.5 V s$^{-1}$ in NKX2-5$^{eGFP/w}$ compared to 2.7 ± 0.2 V s$^{-1}$ in NKX2-5$^{-/-}$ n = 8, p < 0.05; Supplementary Fig. 2d). A further defect in the electrophysiology of NKX2-5$^{-/-}$ cardiomyocytes was demonstrated by a blunted response to the beta adrenoceptor agonist isoprenaline (Fig. 2g). We also determined whether contractile capacity was altered in NKX2-5$^{-/-}$ cardiomyocytes by using bioengineered cardiac organoids[25]. These were generated by placing a single cell suspension of day 15 differentiated cells into a collagen 1 matrix

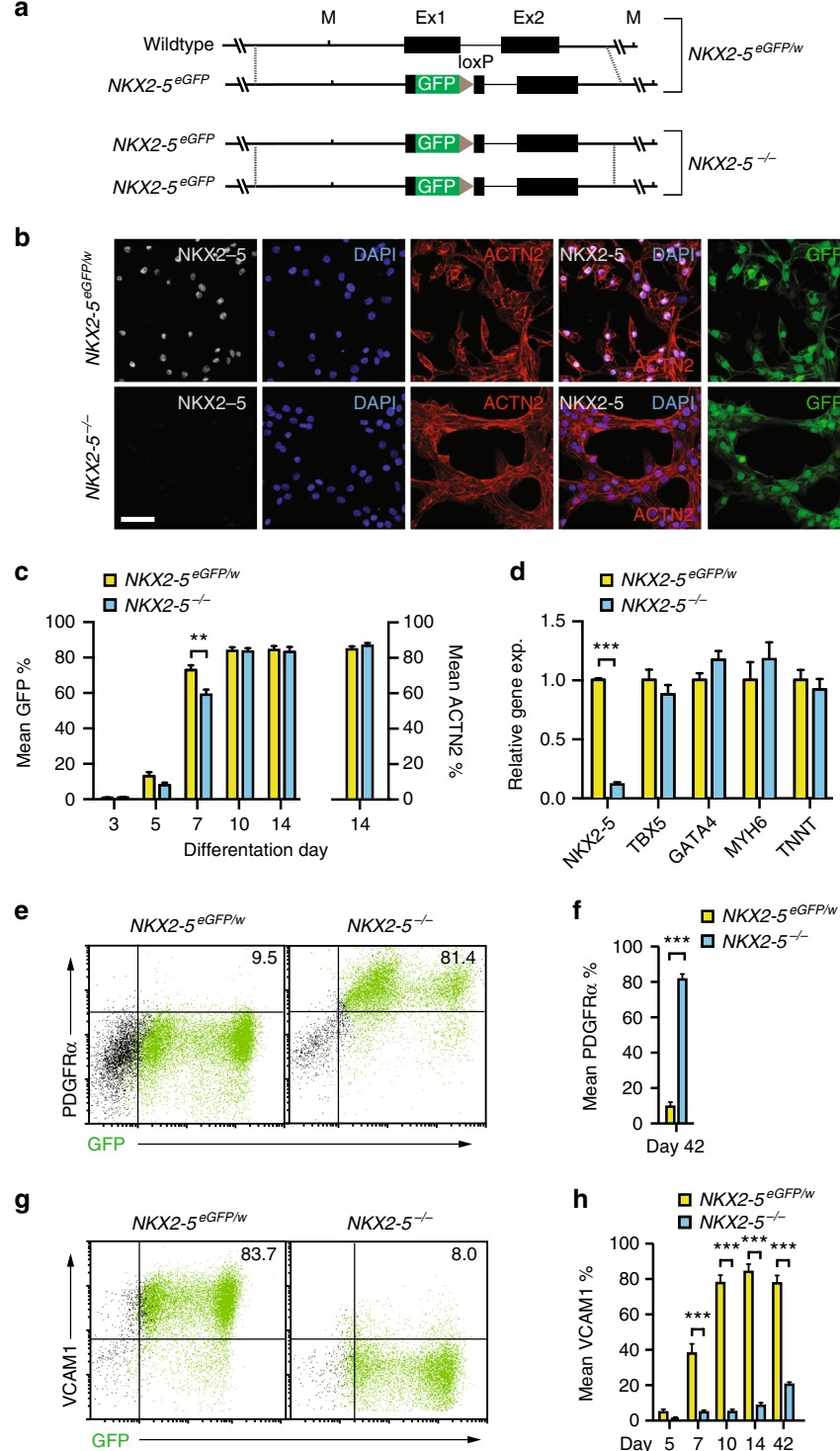

**Fig. 1** *NKX2-5* regulates cardiomyocyte differentiation. **a** Schematic representation of *NKX2-5*eGFP/w and *NKX2-5*−/− (*NKX2-5* null) genotype. **b** Immunofluorescent detection of NKX2-5, ACTN2 and GFP in *NKX2-5*eGFP/w and *NKX2-5*−/− cultures at day 14 of cardiac differentiation. Nuclei counterstained with DAPI. Scale bar = 50 μM. **c** Bar graph quantifying GFP and ACTN2 expression in differentiating *NKX2-5*eGFP/w and *NKX2-5*−/− cultures, as determined by flow cytometry (see Supplementary Fig. 1). Data represent mean ± SEM ($n = 5$). **p < 0.01 (Student's t-test). **d** Q-PCR analysis of *NKX2-5*eGFP/w and *NKX2-5*−/− cultures at day 14 of differentiation. *NKX2-5* null cardiomyocytes show normal expression of characteristic cardiomyocyte markers. Data represent mean ± SEM ($n = 4$). *** $p < 0.001$ (Student's t-test). **e, f** Representative flow cytometry plots (**e**) and bar graph (**f**) of PDGFRα expression in *NKX2-5*eGFP/w and *NKX2-5*−/− cultures at day 42 of differentiation. Numbers on plots are percentage of cells in quadrant. Data represent mean ± SEM ($n = 4$). ***p < 0.001 (Student's t-test). **g, h** Representative flow cytometry plot at day 14 of differentiation (**g**) and bar graph (**h**) of a time course of VCAM1 expression in differentiating *NKX2-5*eGFP/w and *NKX2-5*−/− cultures. Numbers on plots are percentage of cells in quadrant. Data represent mean ± SEM ($n = 4$). ***p < 0.001 (Student's t-test)

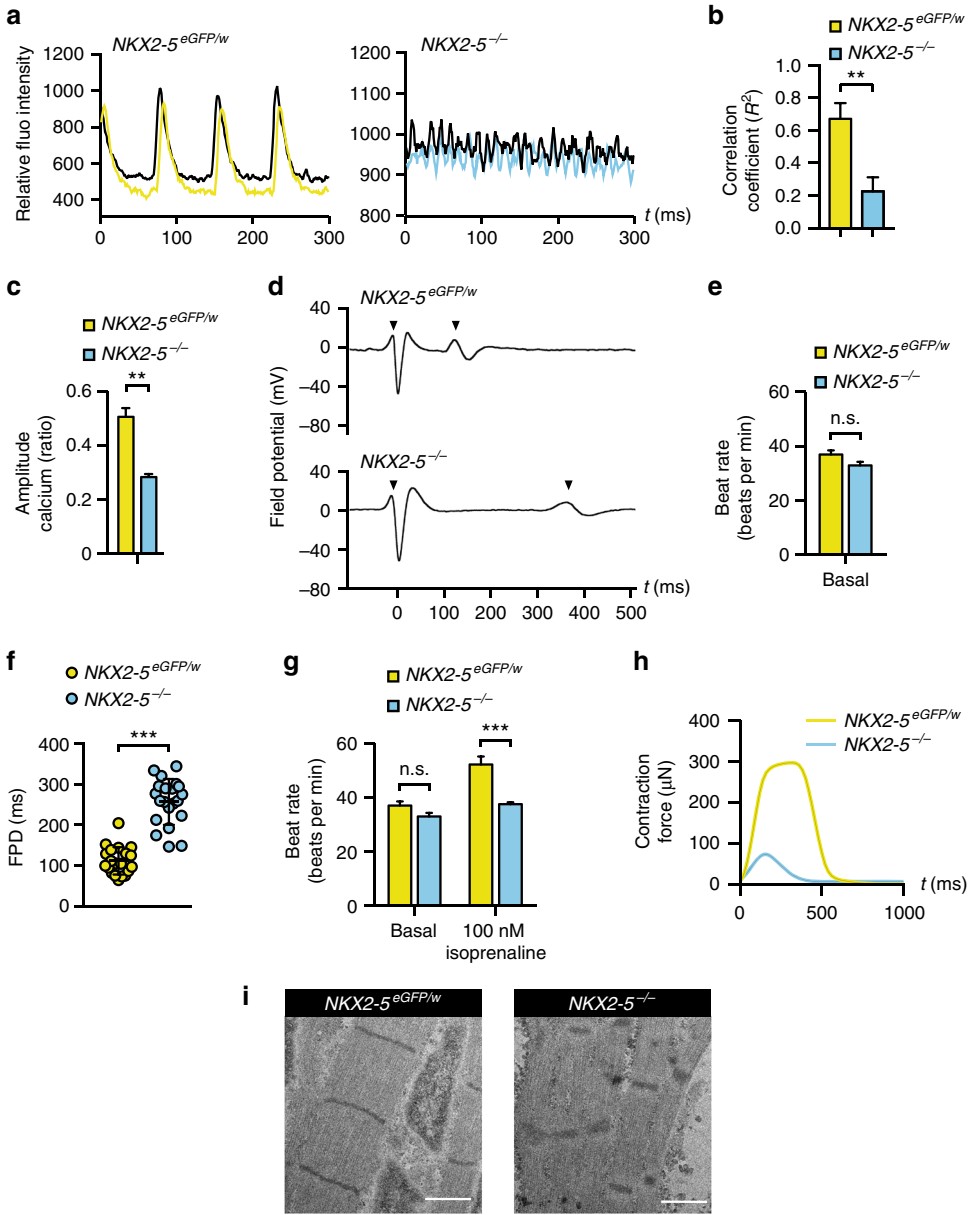

**Fig. 2** Functional profiling demonstrates *NKX2-5⁻/⁻* cardiomyocytes have perturbed electrophysiology and reduced contractile force. **a** Representative graphs showing co-ordination of calcium flux in day 17 cardiomyocyte monolayers derived from *NKX2-5^{eGFP/w}* and *NKX2-5⁻/⁻* hESCs as detected by Fluo4-AM. **b** Bar graph quantifying demonstrating analysis of correlation between calcium imaging signals as derived in **a**. Data represent mean ± SEM (*n* = 6). ** *p* < 0.01 (Student's *t*-test). **c** Bar graphs quantifying calcium amplitude (as a ratio of max to min calcium concentration) during contraction of *NKX2-5^{eGFP/w}* and *NKX2-5⁻/⁻* monolayers at day 14 of differentiation. Data represent mean ± SEM (*n* = 6). ** *p* < 0.01 (Student's *t*-test). **d** Representative traces of MEA extracellular field potentials of cardiomyocyte aggregates derived from *NKX2-5^{eGFP/w}* and *NKX2-5⁻/⁻* cultures at day 14 of differentiation (arrowheads represent start and end of field potential). **e** Bar graph demonstrating *NKX2-5^{eGFP/w}* and *NKX2-5⁻/⁻* cardiomyocyte aggregates have similar rates of contraction at day 14 of differentiation, as determined by MEA. Data represent mean ± SEM (*n* = 13). **f** Dot plots of field potential duration (FPD) of cardiomyocyte aggregates, as derived in **d**. *NKX2-5* null cardiomyocyte aggregates have a prolonged FPD, which is maintained until day 42 of differentiation (Supplementary Fig. 2a). Bars represent mean ± SD (*n* = 20). *** *p* < 0.001 (Student's *t*-test). **g** Bar graphs demonstrating *NKX2-5* null cardiomyocyte aggregates at day 14 of differentiation have an impaired chronotropic response to beta-adrenergic stimulation with isoprenaline, as determined by MEA. Data represent mean ± SEM (*n* = 13). *** *p* < 0.001 (Student's *t*-test). **h** Representative graph of contraction force generated during a single contraction by *NKX2-5^{eGFP/w}* and *NKX2-5⁻/⁻* bioengineered cardiac organoids (see Supplementary Fig. 2f for quantitation). **i** Transmission electron micrographs show that *NKX2-5* null cardiomyocytes have disorganized sarcomeres compared to *NKX2-5^{eGFP/w}* cardiomyocytes (see also Supplementary Fig. 2g). Scale bar = 1 μM

that promotes tissue formation around 2 elastic pillars (Supplementary Fig. 2e, Supplementary Movie 2), a configuration that enables the imposition and measurement of mechanical loading. These cultures were allowed to mature for a further 13 days before analysis. Cardiac organoids from *NKX2-5⁻/⁻* cells

generated significantly reduced contractile force (74 ± 8 μN *n* = 3) compared to *NKX2-5^{eGFP/w}* cells (283 ± 34 μN *n* = 3) (Fig. 2h and Supplementary Fig. 2f). Consistent with impaired bioengineered muscle function the sarcomeres of *NKX2-5* null cardiomyocytes are disorganized (Fig. 2i and Supplementary

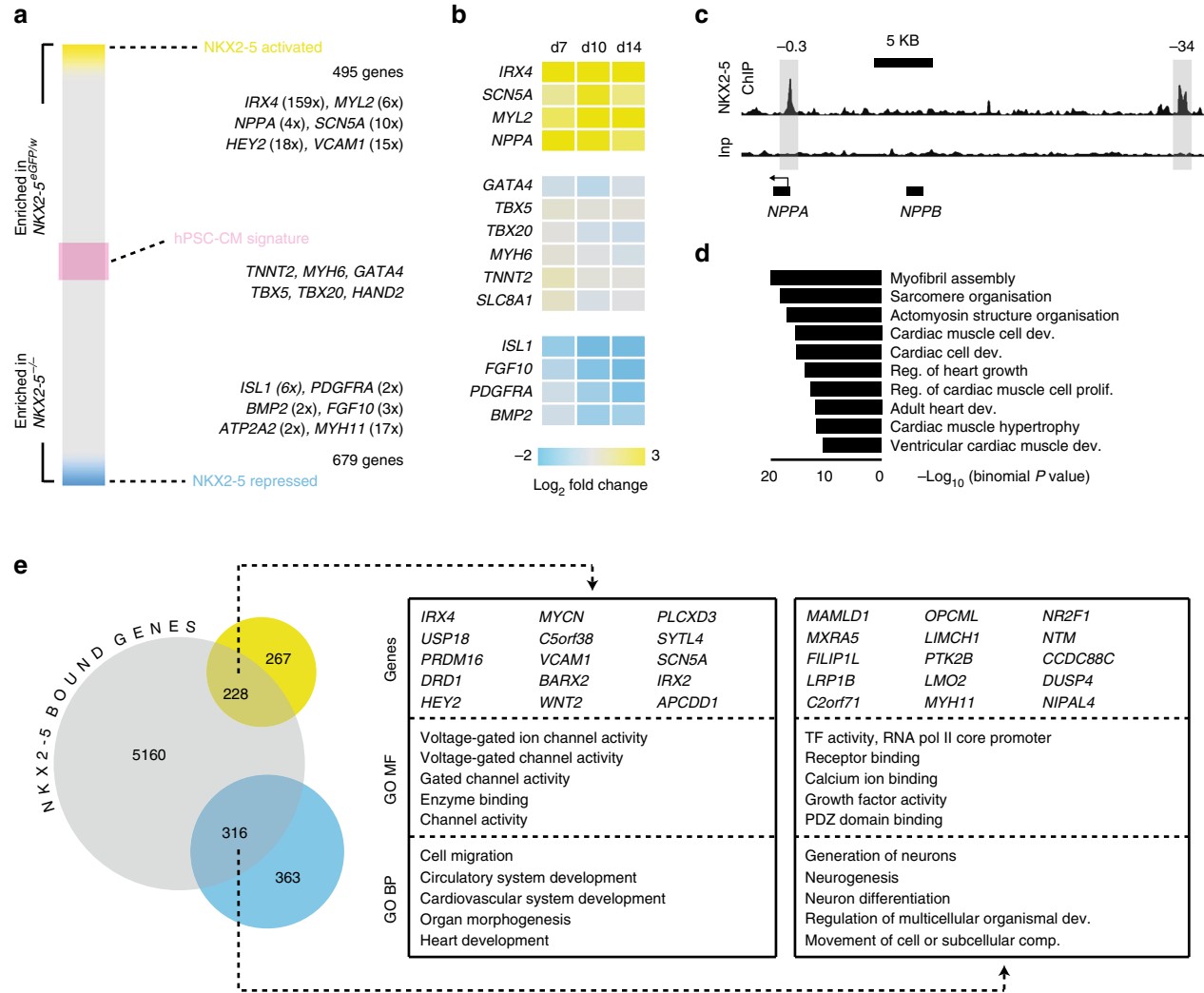

**Fig. 3** Defining the NKX2-5 transcriptional network. **a** Schematic heat map showing differential gene expression between GFP⁺ cells isolated from *NKX2-5^{eGFP/w}* and *NKX2-5^{−/−}* GFP⁺ cultures at day 10 of cardiac differentiation. In *NKX2-5* null GFP⁺ cells, NKX2-5 activated genes (yellow) have reduced expression whereas NKX2-5 repressed genes (blue) have increased expression. Expression of hPSC-CM signature genes (pink) is largely NKX2-5 independent. Numbers represent mean fold change in gene expression (n = 3). **b** Heat map of gene expression in GFP⁺ cells isolated from *NKX2-5^{eGFP/w}* and *NKX2-5^{−/−}* cultures at day 7, 10, or 14 of cardiac differentiation, as determined by Q-PCR. Displayed as mean log₂ fold change between the two genotypes at each time point (n = 4). **c** Representative NKX2-5 ChIP-seq data showing localization of NKX2-5 binding at the *NPPA* locus. Highlighted peaks in NKX2-5 ChIP-seq track denote conserved NKX2-5 binding regions at −0.3 kb and −34 kb from transcriptional start site enriched after chromatin immunoprecipitation with NKX2-5. Inp = input chromatin. **d** Most represented GO biological process terms returned when the closest genes to NKX2-5 binding sites were analyzed. This data shows NKX2-5 binds near genes involved in heart development and cardiomyocyte function. **e** Venn diagram outlining overlap between genes positively (yellow) and negatively (blue) regulated by NKX2-5, and NKX2-5 bound genomic regions (gray). Boxes contain top 15 differentially regulated genes with proximal NKX2-5 binding sites and top 5 GO terms (MF = molecular function, BP = biological process) from the genes within the overlapping regions of the Venn diagram

Fig. 2g). Thus, *NKX2-5* null cardiomyocytes displayed intrinsic defects in force generation and action potential characteristics.

**Defining the human *NKX2-5* genetic network.** To understand how human *NKX2-5* regulates myocardial differentiation, we defined the NKX2-5 genetic network by combining gene expression and chromatin immunoprecipitation sequencing (ChIP-seq) analysis. Expression profiling of day 10 differentiated cells from both genotypes, enriched for cardiomyocyte lineage committed cells on the basis of high GFP expression[18–20], identified 1174 differentially regulated genes (≥2 fold change, adj. *p* value < 0.05; Fig. 3a, Supplementary Data 1). As expected from the contractile nature of *NKX2-5^{−/−}* cultures, the majority of

genes within a defined hPSC-CM signature[26] were not differentially expressed in *NKX2-5^{−/−}* cardiomyocytes (63/99; Fig. 3a). The 495 genes more abundant in *NKX2-5^{eGFP/w}* cultures included the known *NKX2-5* target genes *NPPA* and *IRX4*[27]. There was reduced expression of a number of ventricular specific markers including *IRX4*, *HAND1* and *MYL2* in *NKX2-5* knockout cultures (Fig. 3a, Supplementary Data 1), consistent with the predominant ventricular-like cardiomyocytes generated in monolayer cardiac differentiations[28]. Six hundred and seventy nine genes were more highly transcribed in *NKX2-5^{−/−}* cardiomyocytes implying that *NKX2-5* was required to repress these genes during differentiation. These included known markers of the cardiac progenitor cells, such as, *ISL1*, *PDGFRA*, *BMP2* and *FGF10* (Fig. 3a), that were previously found to be upregulated in the hearts of *Nkx2-5*

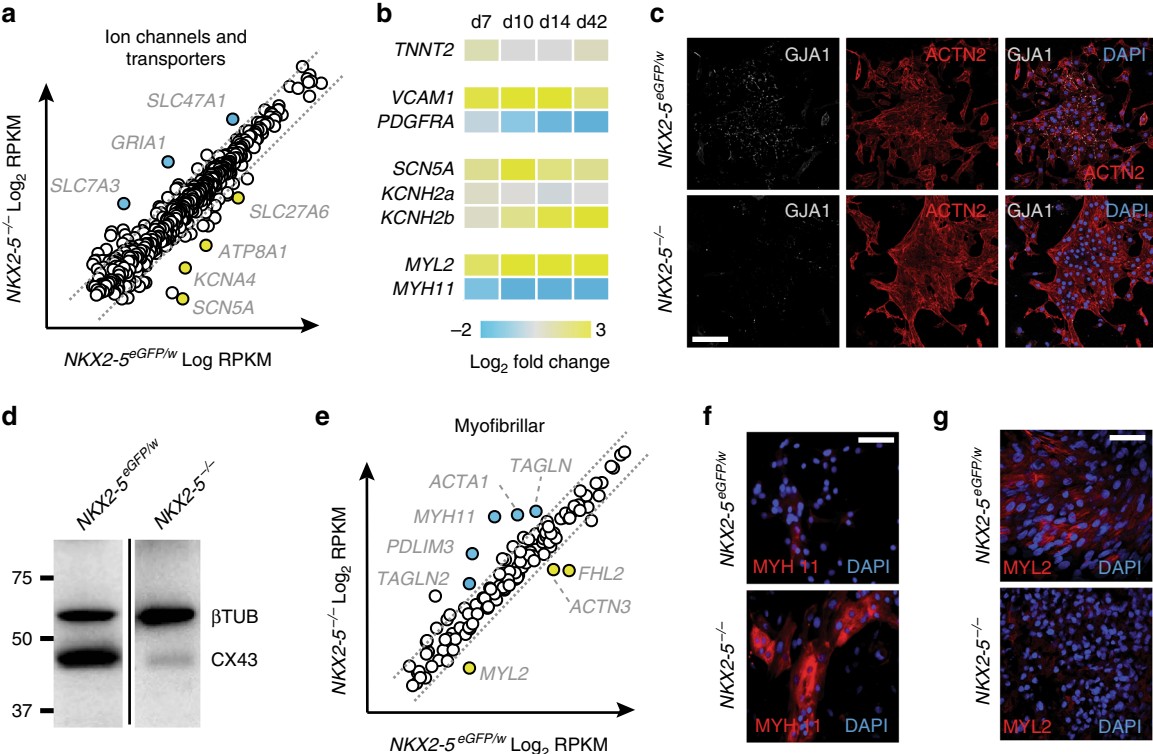

**Fig. 4** Deletion of NKX2-5 disrupts both electrical and mechanical gene networks in cardiomyocytes. **a** Dot plot representation of RNA-seq absolute gene expression (log$_2$ RPKM values) for a reported list of ion channel and transporter genes. Dotted lines mark 2-fold differential expression level. **b** Heat map of gene expression (Q-PCR) in GFP$^+$ cells isolated from *NKX2-5$^{eGFP/w}$* and *NKX2-5$^{-/-}$* cultures at day 7, 10, 14, or 42 of cardiac differentiation. Displayed as mean log$_2$ fold change between the two genotypes at each time point (*n* = 4). **c** Immunocytochemistry analysis of GJA1 (CX43) and ACTN2 expression in cardiomyocytes derived from *NKX2-5$^{eGFP/w}$* and *NKX2-5$^{-/-}$* cells at day 42 of differentiation. Scale bar = 100 μm. **d** Western blot detection of GJA1 in *NKX2-5$^{eGFP/w}$* and *NKX2-5$^{-/-}$* cultures confirms reduction in GJA1 observed in **h**. Size markers in kDa are indicated to the left of the blot. **e** Dot plot representation of RNA-seq absolute gene expression (log$_2$ RPKM values) for myofibrillar genes. Dotted lines mark 2-fold differential expression level. **f** Immunofluorescent detection of MYH11 (smooth muscle myosin heavy chain) in cardiomyocytes derived from *NKX2-5$^{eGFP/w}$* and *NKX2-5$^{-/-}$* cells at day 14 of differentiation. Nuclei are counterstained with DAPI. Scale bar = 50 μm. **g** Immunofluorescent detection of MYL2 (Myosin light chain 2 v) in cardiac organoids generated from *NKX2-5$^{eGFP/w}$* and *NKX2-5$^{-/-}$* cells. Nuclei are counterstained with DAPI. Scale bar = 50 μm

null mice[7]. Q-PCR demonstrated that the altered gene expression profile of *NKX2-5* null cardiomyocytes is maintained during cardiac differentiation (Fig. 3b). Furthermore, expression of the *NKX2-5*-dependant genes *HEY2*, *IRX4*, *NPPA*, *MYL2* and *VCAM1* is reduced in H9 *NKX2-5* knockout cardiomyocytes (Supplementary Fig. 3a). In addition, transcripts of the progenitor markers *ISL1*, *FGF10* and *BMP2* are upregulated in H9 *NKX2-5* null cardiomyocytes (Supplementary Fig. 3a). Heterozygosity for *NKX2-5* did not alter *IRX4*, *HEY2*, *NPPA* or *VCAM1* expression (Supplementary Fig. 3b) consistent with the similar levels of NKX2-5 protein observed (Supplementary Fig. 1f). Collectively, these data provide molecular evidence supporting the hypothesis that *NKX2-5* is required for the progression of cardiomyocytes into specialized ventricular phenotype, already implied by both the failure to activate VCAM1 and the persistence of PDGFRα cells in *NKX2-5$^{-/-}$* cultures (Fig. 1e–h and Supplementary Fig. 1l, m).

ChIP-seq detected NKX2-5 bound at 5704 sites across the genome. Fidelity of the data set was supported by enrichment of NKX2-5 binding at highly conserved elements upstream of *NPPA* (Fig. 3c)[29] and at genes involved in cardiac muscle development and function (Fig. 3d). In addition, the NKX2-5 binding motif (known as an NK2 element or NKE,) was overrepresented in the sequences bound by NKX2-5[12,14] and binding motifs of other cardiac transcription factor families (e.g., GATA, T-Box) were found within NKX2-5 bound sequences (Supplementary Fig. 3c).

NKX2-5 binding sites displayed a bi-modal distribution relative to transcriptional start sites, with most found >50 Kb from start sites, suggesting that NKX2-5 does not often occupy proximal promoter regions (Supplementary Fig. 3d). NKX2-5 was found at the VCAM1 locus, which, when combined with the differential expression of this myocardial commitment marker, suggests VCAM1 may be a direct NKX2-5 regulatory target (Fig. 3a, e and Supplementary Fig. 3e,f). Conversely, the absence of proximal NKX2-5 binding at the *PDGFRA* locus suggests that any regulatory relationship between NKX2-5 and *PDGFRA* is reliant upon putative NKX2-5-bound enhancers located over 250 kb from the locus (Supplementary Fig. 3e).

Intersection of NKX2-5 binding associated genes (closest gene, GREAT database) with *NKX2-5* dependent genes (495 activated and 679 repressed genes) identified 544 potential direct transcriptional targets of NKX2-5 (Fig. 3e and Supplementary Data 1, 2). Gene ontology (GO) analysis of NKX2-5 bound gene subsets identified GO Biological Process terms correlated with NKX2-5 activated genes that were closely aligned with heart development whilst the GO Molecular Function profile included terms associated with gated channel activity (Fig. 3e and Supplementary Data 3, 4). Further investigation of ion channel and transporter genes identified a subset with altered expression profiles in *NKX2-5* null cells (Fig. 4a). Q-PCR during a time course of differentiation (day 7 to 42) on a subset of genes including ion channels (*SCN5A*, *KCNH2b*), cell surface markers

(*VCAM1*, *PDGFRA*) and myofilament genes (*MYL2*, *MYH11*) demonstrated that differential expression for these genes was maintained throughout differentiation (Fig. 4b). *SCN5A*, required for Nav1.5 channel activity and depolarization of hPSC-CMs[30], was expressed at a lower level in *NKX2-5*[−/−] cells, and NKX2-5 was bound at this locus. *KCNH2b* (HERG1b) is critical for cardiac repolarization[31] and was down-regulated in *NKX2-5*[−/−] cells, whereas expression of the longer isoform, *KCNH2a*, was unperturbed (Fig. 4b), suggesting that *NKX2-5* may only directly regulate the shorter 2b isoform[32]. In support of this notion, NKX2-5 was found bound at a putative promoter region of *KCHN2b* (Supplementary Fig. 4a), within an intron of the

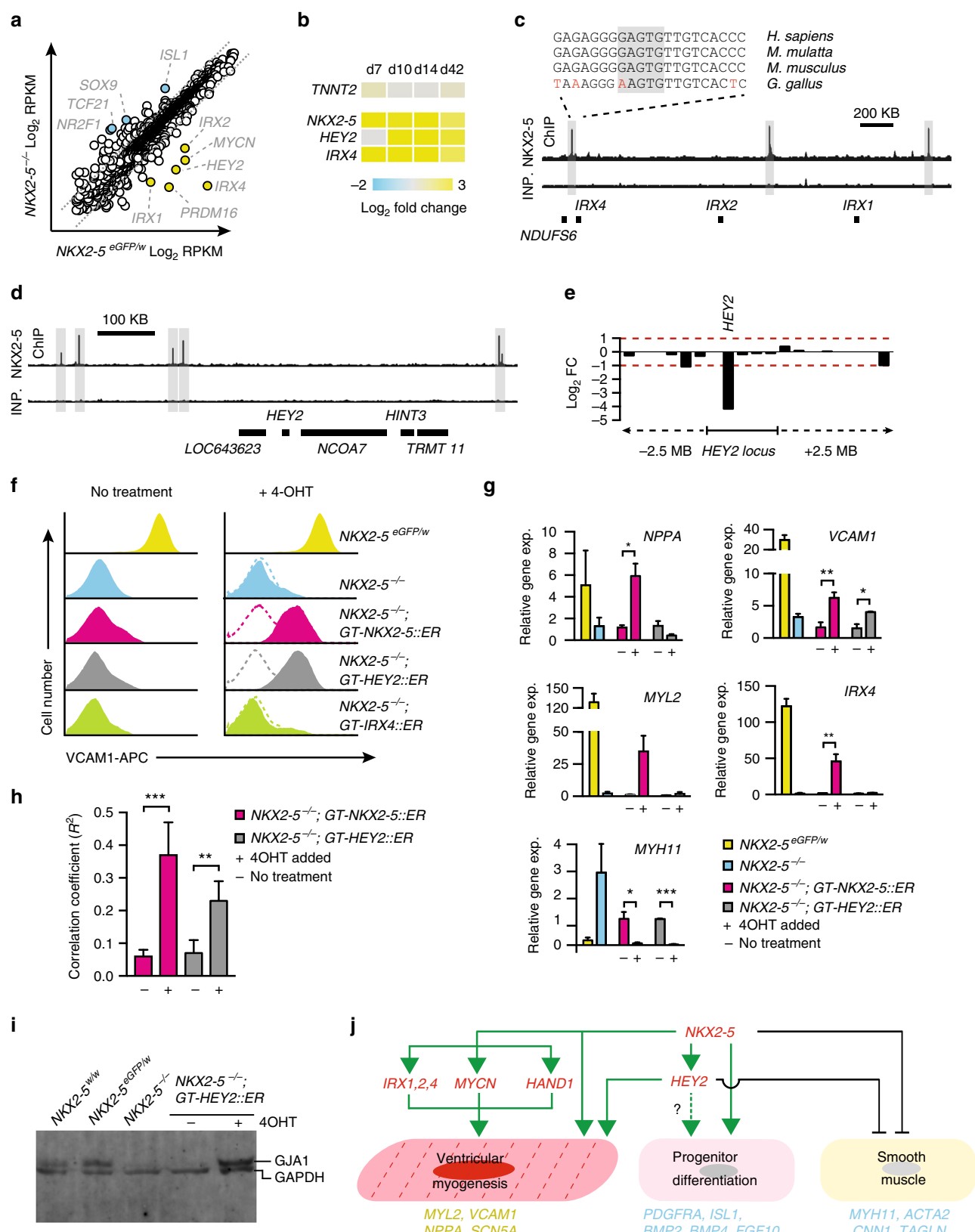

*KCHN2a* transcript. Altered ion channel and transporter gene expression led us to examine expression of connexins, which are important for conduction of electrical signals through gap junctions[33]. GJA1 (Connexin 43) showed expected punctate localization along the periphery of *NKX-5^eGFP/w* cells, a pattern that was lost in *NKX-5* null cardiomyocytes (Fig. 4c). The failure of GJA1 to be robustly incorporated into gap junctions may have reflected the dramatically reduced level of GJA1 protein in *NKX-5^{−/−}* cultures (Fig. 4d). We speculate that the combined effect of improper gap junction formation and altered ion channel and transporter gene expression was most likely responsible for the asynchronous contractility observed in *NKX-5^{−/−}* cultures (Fig. 2a).

As well as electrophysiological abnormalities, the *NKX-5* null cardiac cultures had impaired contractile force (Fig. 2h and Supplementary Fig. 2f). Profiling of myofibrillar components and smooth muscle associated genes revealed that *NKX-5^{−/−}* cardiac cells expressed higher levels of the smooth muscle genes *CNN1*, *MYH11*, *ACTA2*, *TAGLN*, and *CALD1* than *NKX-5^eGFP/w* (Fig. 4e and Supplementary Fig. 4b). Furthermore, NKX-5 was bound at the *MYH11* and *TAGLN* loci and at a series of other smooth muscle proteins (Supplementary Data 2), suggesting NKX-5 normally represses these genes. Supporting this hypothesis, MYH11 protein was found at higher levels in *NKX-5^{−/−}* cardiomyocytes (Fig. 4f). Conversely, *MYL2* transcription and protein levels were reduced in *NKX-5^{−/−}* cardiomyocytes (Figs. 3a and 4e, g), further underlining the requirement for *NKX-5* for cardiomyogenesis. Together, these data suggest that progression to a ventricular cardiac phenotype is blocked in *NKX-5^{−/−}* cardiomyocytes and NKX-5 is required to repress the ancestral smooth muscle genetic program. Alternatively or additionally, it also possible that in the absence of NKX-5, heart progenitor cells with cardiomyocyte and smooth muscle potential[24,34] may preferentially adopt a smooth muscle fate.

Finally, *NKX-5* has a conserved role regulating the genetic program of transient embryological structures such as the second heart field, atrioventricular canal and outflow tract[6]. Whilst 2D differentiation lacks the spatiotemporal signaling and patterning driving cardiogenesis in the embryo, a number of important developmental genes were nevertheless dysregulated in *NKX-5^{−/−}* cultures. Expression of *FGF10*, *ISL1* and *MEF2C* and the atrioventricular canal markers *SOX4*, *SOX9* and *TWIST1* were increased in *NKX-5^{−/−}* cardiomyocytes (Supplementary Fig. 4c, d). Further, binding of NKX-5 at the *FGF10*, *ISL1*, *SOX4* and *TWIST1* loci (Supplementary Fig. 4e) suggested direct negative regulation for these genes. In addition, BMP2, which is known to potentiate second heart field expansion in *Nkx-5^{−/−}* mice[7], was expressed more highly in *NKX-5^{−/−}* cardiomyocytes (Fig. 3b and Supplementary Fig. 4f). This data shows that both important developmental genes and markers of specialized non-myocyte

lineages are dysregulated in *NKX-5* null cells and the presence of NKX-5 at these loci supports an important, conserved role for human NKX-5 in these developmental processes and cell types[7,35].

**HEY2 mediates NKX-5 activity.** To determine the network of transcription factors controlled by NKX-5, we compared the expression of all predicted human transcription factors[36] between *NKX-5^eGFP/w* and *NKX-5^{−/−}* cardiomyocytes (Fig. 5a). Expression of most cardiac GRN members, including *GATA4* and *TBX5*, was not dependent on NKX-5 (Fig. 3b). The most differentially expressed NKX-5-dependent transcription factors were *MYCN*, *PRDM16*, *HEY2* and the *IRX1/2/4* cluster (Fig. 5a). Each of these genes has proximal NKX-5 binding sites (Fig. 5c, d and Supplementary Data 2) and all are required for normal ventricular development and function[37–39]. *MYCN* and *IRX4* have been identified as NKX-5-dependant genes in the mouse[27,40]. Further, the *IRX4* and *HEY2* transcription factors are also dysregulated in H9 *NKX-5^eGFP/del* cardiomyocytes (Supplementary Fig 3a). Since the majority of cardiomyocytes obtained in monolayer differentiations of wildtype hPSCs display an early embryonic ventricular phenotype (by action potential and gene expression signature) we focused on *HEY2* and *IRX4* as they have known roles in murine ventricular myogenesis[27,39,41–47]. Further, *Hey2* is a downstream target of the Notch pathway, which is known to be in important for ventricular muscle development, and is enriched in the compact myocardial layer[9,48].

All members of the *IRX1/2/4* cluster, but not the duplicated *IRX3/5/6* cluster, were differentially expressed between *NKX-5^eGFP/w* and *NKX-5^{−/−}* cardiomyocytes (Supplementary Fig. 5a, b). With the exception of slightly reduced levels of *IRX3* expression in *NKX-5^{−/−}* cardiomyocytes, *IRX3/5/6* cluster transcription did not vary greatly between the two lines (Supplementary Fig. 5a). Differential *IRX4* expression was observed throughout the course of cardiac differentiation (Fig. 5b) and IRX4 protein levels were reduced in *NKX-5^{−/−}* cardiomyocytes (Supplementary Fig. 5d). The *IRX1/2/4* cluster is likely a direct target of NKX-5, as NKX-5 is bound at multiple locations across the genomic region (Fig. 5c). In addition, the NKX-5 binding sites are highly conserved between species, indicating they likely mark functional enhancers regulating cardiac expression of the *IRX1/2/4* locus (Fig. 5c).

The *HEY2* locus is flanked by four upstream (−410 kb, −380 kb, −210 kb, and −190 kb) and two downstream (+375 kb and +370 kb) NKX-5 binding sites (Fig. 5d). Although there are four other genes in the vicinity, *HEY2* is the only one within 5 Mb that is differentially expressed in the absence of NKX-5 (Fig. 5e) and is the only HES/HEY family member differentially expressed

**Fig. 5** HEY2 is a key downstream transcriptional mediator of *NKX-5*. **a** Dot plot representation of RNA-seq absolute gene expression (log$_2$ RPKM values) for FANTOM5 predicted transcription factors. Dotted line marks 2 fold differential expression level. **b** Heat map of gene expression in GFP$^+$ cells isolated from *NKX-5^eGFP/w* and *NKX-5^{−/−}* cultures at day 7, 10 or 14 of cardiac differentiation, as determined by Q-PCR. Displayed as mean log$_2$ fold change between the two genotypes at each time point ($n = 4$). **c, d** Schematics of NKX-5 ChIP-seq data showing the *IRX1/2/4* cluster (**c**) and *HEY2* locus (**d**) with regions bound by NKX-5 highlighted. The IRX4 proximal NKX-5 bound region is highly conserved. Inp. = input chromatin. **e** Differential expression of genes 2.5 Mbp up or downstream of the *HEY2* locus in **d**. This data shows *HEY2* is the only differentially expressed gene in this chromosomal region. Red dashed line marks 2 fold (adj. $p$ value < 0.05) gene expression difference between genotypes. **f** Histograms of flow cytometry analysis of VCAM1 in untreated (No treatment) or induced (+4-OHT) *NKX-5^{−/−}* GAPTrap (GT) lines. Both *GT-NKX-5::ER* and *GT-HEY::ER* restore VCAM1 expression ($n = 4$). **g** Gene expression profiling of genetic rescue via the modified GAPTrap loci, *GT-NKX-5::ER*, *GT-HEY::ER* and *GT-IRX4::ER*, as determined by Q-PCR ($n = 3$). * $p < 0.05$, ** $p < 0.01$, *** $p < 0.001$ (Student's $t$-test). **h** Correlation coefficient between contractile areas improves when both *NKX-5* and *HEY2* are induced ($n = 3$, scored blind to genotype). ** $p < 0.01$, *** $p < 0.001$ (Student's $t$-test). **i** Western blot showing restoration of GJA1 (connexin 43) levels by HEY2 and that wildtype (HES3) and *NKX-5^eGFP/w* GJA1 levels are comparable. **j** Network model of NKX-5 regulated genes and their potential roles in regulating ventricular myogenesis, progenitor differentiation and smooth muscle differentiation. Representative genes with altered expression (yellow text activated genes, blue repressed genes) in *NKX-5* null cultures are shown below each process

between $NKX2\text{-}5^{eGFP/w}$ and $NKX2\text{-}5^{-/-}$ cardiomyocytes (Supplementary Fig. 5c). In addition, *HEY2* expression is dramatically reduced in *NKX2-5* null cultures throughout cardiac differentiation (Fig. 5b) and HEY2 protein levels are reduced in $NKX2\text{-}5^{-/-}$ compared to $NKX2\text{-}5^{eGFP/w}$ cardiomyocytes (Supplementary Fig. 5e). Furthermore, NKX2-5, in collaboration with established transcriptional co-factors GATA4 and TBX20[49], is able to transactivate both proximal 5′ (−190 kb) and 3′ (+379 kb) putative enhancer elements in HEK 293 T cells (Supplementary Fig. 5f). In some contexts *Hey2* is induced by BMP and TGFβ signaling[15] and expression of components of both these pathways is reduced in *NKX2-5* null cardiomyocytes (Supplementary Fig. 4f), which may lead to a decrease in *HEY2*. However, *HEY2* transcript levels were not reduced when differentiating $NKX2\text{-}5^{eGFP/w}$ cardiomyocytes were exposed to the BMP antagonist DHM1 (Supplementary Fig. 5g). Taken together these data support the hypothesis that HEY2 is directly regulated by NKX2-5.

In order to determine the role of HEY2 and IRX4 in the *NKX2-5* gene regulatory network we used the GAPTrap system (GT)[50] to express *NKX2-5*, *IRX4* and *HEY2* fused to a mutated estrogen receptor domain (ER) that permits temporal induction of protein activity by the addition of the estrogen analog 4OHT (Supplementary Fig. 5h, i and ref. [51]). Using VCAM1, which marks committed cardiomyocytes[16,18,52] as a readout of phenotypic rescue, we demonstrated that induction of NKX2-5 expression in $NKX2\text{-}5^{-/-}$;GT-NKX2-5::ER cells was permissive for continued cardiac differentiation (Fig. 5f). Induction of IRX4 activity did not restore the cardiomyogenic program in $NKX2\text{-}5^{-/-}$ cells, as assayed by VCAM1 expression (Fig. 5f). However, GAPTrap based expression of *HEY2* restored VCAM1 expression to a level similar to that observed in both $NKX2\text{-}5^{eGFP/w}$ and $NKX2\text{-}5^{-/-}$; *GT-NKX2-5::ER* control cultures (Fig. 5f and Supplementary Fig. 5j). Q-PCR analysis of *NPPA* expression demonstrated that both NKX2-5::ER and HEY2::ER fusion proteins retained transcriptional activity. *NPPA* is positively regulated by NKX2-5 and negatively regulated by HEY2 in the developing mouse heart[29], a relationship reproduced in vitro when NKX2-5 and HEY2 function were induced during the differentiation of NKX2-5 null cells (Fig. 5g). Gene expression analysis also confirmed partial restoration of *VCAM1* mRNA levels by both NKX2-5 and HEY2 in $NKX2\text{-}5^{-/-}$ cells, and repression of the smooth muscle myofilament gene *MYH11*, which was strongly upregulated in $NKX2\text{-}5^{-/-}$ cultures (Fig. 5g). However, HEY2::ER only rescued a subset of the NKX2-5 dependent transcriptome, for example HEY2::ER expression did not result in activation of *MYL2* and *IRX4* (Fig. 5g). Finally, in both $NKX2\text{-}5^{-/-}$;*GT-NKX2-5::ER* and $NKX2\text{-}5^{-/-}$;*GT-HEY2::ER* cultures contractile synchronicity was restored to a similar level (Fig. 5h, Supplementary Movie 3) and GJA1 protein levels were restored in HEY2::ER rescued cultures to levels comparable to both $NKX2\text{-}5^{eGFP/w}$ and wildtype cardiomyocytes (Fig. 5i). Thus, HEY2 is able to rescue important aspects of the *NKX2-5* null phenotype. Taken together these data support the hypothesis that *HEY2* is one of the critical mediators of the *NKX2-5*-dependent transcriptional network that guides cardiomyocyte differentiation (Fig. 5j).

## Discussion

*NKX2-5* is essential to establish the transcriptional program for ventricular muscle development. HPSC-CMs derived from $NKX2\text{-}5^{-/-}$ cells, failed to activate VCAM1 and inappropriately maintained expression of the progenitor marker PDGFRA. In the mouse, VCAM1 mediates ventricular myocardial development through interactions with α−4-integrin presented on the epicardium[53,54]. Furthermore, gene expression and genomic binding

profiling demonstrated dysregulation of the ventricular myogenic program and key progenitor genes, with higher expression of smooth muscle, second heart field and atrioventricular genes and loss of normal ion channel gene expression in $NKX2\text{-}5^{-/-}$ derived cardiomyocytes. These changes subsequently manifest as reduced contractile force and asynchronous contraction of cardiac sheets in the *NKX2-5* mutant cells.

We identified *MYCN*, *PRDM16*, *HEY2*, and the *IRX1/2/4* cluster as candidate transcription factors required for normal ventricular development that might mediate *NKX2-5* function, and focused on the role of the *IRX1/2/4* cluster and *HEY2* gene as downstream of *NKX2-5*. Despite its early expression in ventricular myocardium[39], enforced expression of *IRX4* failed to upregulate VCAM1 in differentiating $NKX2\text{-}5^{-/-}$ cells, and did not normalize the asynchronous contractions patterns that were a hallmark of the $NKX2\text{-}5^{-/-}$ cardiomyocytes. Conversely, induced expression of *HEY2* partially rescued the *NKX2-5* phenotype, including restoring GJA1 levels, without up regulating *IRX4*. *Hey2* regulates ventricular myocardial development, in part by suppressing the atrial gene expression program and has recently been found to be more highly expressed in the compact myocardium[9,43,44,55]. *Hey2* knockout mice have severe ventricular septal defects and cardiac valve malformations, which result in neo-natal death[56,57]. Further, the *Hey* genes (*Hey2*, *Hey1*, and *HeyL*) control atrioventricular canal formation and subsequent valve formation and septation by regulating epithelial to mesenchymal transition (reviewed in[15]). In humans, *HEY2* mutations are associated with Brugada syndrome, a ventricular arrhythmia, which can cause sudden death[58]. Our data suggest that *HEY2* is a key component of the NKX2-5 transcriptional network. This finding is consistent with the overlapping phenotypes of conduction system abnormalities in individuals with pathogenic *NKX2-5* and *HEY2* mutations[5,58,59].

Several lines of evidence suggest the NKX2-5-HEY2 regulatory relationship is direct. First, *HEY2* expression in cardiomyocytes is dependent on NKX2-5. Second, while NKX2-5 is bound at DNA elements some distance from the *HEY2* translational start site, *HEY2* is the only gene within 5 Mbp that has altered expression in NKX2-5 null cells. Third, NKX2-5, in the presence of known co-factors GATA4 and TBX20, was able to transactivate two of these *HEY2* regulatory elements in a heterologous system. Further, inhibition of the BMP signaling pathway does not alter *HEY2* levels suggesting that in cardiomyocytes *HEY2* expression is not regulated by a *BMP* regulatory axis. It is likely that *HEY2* regulation is multifactorial and complex. In this context, it is interesting to note increased expression of *NR2F2* (COUP-TFII), a known repressor of *HEY2*[46], in *NKX2-5* null cardiomyocytes. Thus, *HEY2* regulation by NKX2-5 may include an indirect component through COUP-TF-dependent repression. It is clear that *HEY2* is an important *NKX2-5*-dependant factor for human ventricular muscle differentiation and, based on findings in the mouse, may drive compact myocardium development[9]. Furthermore, these findings suggest that the HEY2 and NKX2-5 downstream targets coordinate synchronicity of excitation/contraction coupling which is necessary to drive heart function during early human embryogenesis.

In summary, our study demonstrates the utility of hPSCs for the molecular dissection of human cardiac development and sheds light on the *NKX2-5* dependent regulatory axis that drives cardiogenesis. These results provide a framework for further analysis of the function and interdependence of the network of NKX2-5 downstream transcription factors in early human cardiac development.

## Methods

**Genetic manipulation of hESC lines**. The NKX2-5 locus was genetically modified and correctly targeted clones identified by PCR and Southern blotting using

standard protocols[16]. Modification of the GAPDH locus and identification of correctly targeted clones was performed using established methods[50]. CRISPR/Cas9 genome editing was used to delete the coding sequence of NKX2-5. Briefly, synthetic oligonucleotides containing the desired NKX2-5 protospacer sequence (5′ CCATGTTCCCCAGCCCT and 5′ GACCGATCCCACCTCAAC) and sequence overhangs compatible to the BbsI were annealed and the duplex cloned into the BbsI site of the vector pSpCas9(BB)-2A-GFP vector (PX458; Addgene Plasmid #48138)[60]. Subsequently, H9 cells[61] in which one allele had been targeted with sequences encoding eGFP, H9 NKX2-5$^{eGFP/w}$[16], were electroporated with the plasmid, and GFP-expressing single cells were isolated by FACS after 2–5 days using a BD Influx cell sorter[62]. Individual GFP-expressing clones were expanded and screened by PCR (NKX2-5 Fwd 5′ TTGTGCTCAGCGCTACCTGCTGC and NKX2-5 rev 5′ GGGGACAGCTAAGACACCAGG) to identify clones with modified alleles. The mutant alleles were confirmed by sequencing of the PCR products and pluripotency of the H9 NKX2-5$^{eGFP/del}$ was confirmed by expression of pluripotent stem cell markers (ECAD, SSEA-4, TRA160, CD9) and differentiation to mesodermal and endodermal lineages. Genomic integrity of selected genetically modified lines was assessed either using the Illumina HumanCytoSNP-12 v2.1 array at the Victorian Clinical Genetics Service, Royal Children's Hospital (Melbourne) or by karyotyping by the Cytogenetics Department at the Monash Medical Centre with a total of 20 metaphase chromosome spreads examined for each line. H9 cells were obtained from WiCell (WA09)[61] and HES3 human embryonic stem cells lines were isolated and characterized by Richards and colleagues[63]. Human ESC work was approved by the Monash Medical Centre and Royal Children's Hospital Human Research Ethics Committees.

**Cell culture and cardiac differentiation.** All cell culture reagents purchased from Thermo Fisher unless stated. HES3 and derivative NKX2-5 targeted cell lines were cultured on 75 cm$^2$ tissue culture flasks and passaged using TrypLE Select as described previously[16]. To induce differentiation, hESCs were harvested using TrypLE Select and seeded on Geltrex coated cell culture plates at 2.5 × 10$^5$ cells/cm$^2$ in basal differentiation media consisting of RPMI (Thermo 61870), B27 minus vitamin A (Thermo 12587) and 50 µg/ml ascorbic acid (Sigma), further supplemented with 10 µM CHIR99021 (Tocris Bioscience) and 80 ng/ml Activin A (Peprotech). At 24 and 96 h following induction of differentiation, media was changed to basal differentiation media supplemented with 5 µM IWR-1 (Sigma), and from day 5, differentiating cultures were maintained in basal differentiation media only.

**Flow cytometry.** Flow cytometry analysis and sorting of lives cells was performed for GFP, VCAM1 (diluted 1:100, biotin conjugated Abcam ab7224) detected with APC-Streptavidin conjugated secondary (1:100, Biolegend), and PDGFRA (BD Biosciences, 556001) detected with PE/Cy7 conjugated secondary (Biolegend, 405315), as described previously[16,18,64]. Pluripotency markers used were ECAD (ThermoFisher Scientific, MA1-10192) detected with APC conjugated secondary (1 in 100), EpCAM-PE (Biolegend, 324205, diluted 1:100), CD9-FITC (BD Biosciences, 341646, diluted 1:100) and SSEA4-APC (Biolegend, 330418, diluted 1:100) were detected as For intracellular flow cytometry, cells were harvested with TrypLE Select, fixed in 4% paraformaldehyde for 15 min at room temperature, blocked and permeablised in block buffer consisting of 1 × Perm/Wash Buffer (BD) and 4% goat serum (Sigma) for 15 min at 4 °C. Cells were then incubated with ACTN2 antibody (Sigma, A7811, diluted 1:100) for 1 h at 4 °C and then Alexa Fluor 647 conjugated secondary (ThermoFisher Scientific, A-21235, diluted 1:1000) for 1 h at 4 °C. Collection of flow cytometric data was performed using BD Fortessa™ analyser and analyzed with FlowLogic software (Inivai Scientific). Cell sorting was done using FACS Diva™ and BD Influx™ cell sorters (BD Biosciences).

**Immunofluorescence.** Immunofluorescence was performed on cells seeded onto Geltrex coated optical tissue culture treated 96 well plates (Greiner 665090). Cells were fixed in 4% PFA in PBS for 15 min, then blocked in block buffer consisting of PBS, 1 × Perm/Wash Buffer (BD), 0.1 mg/ml human IgG (Sigma) and 4% goat serum (Sigma) for 15 min at 4 °C. All antibodies were diluted in PBS with 1 × Perm/Wash Buffer. Primary antibody staining was performed overnight at 4 °C for NKX2-5 (Santa Cruz sc-14033, diluted 1:1000), ACTN2 (Sigma A7811, diluted, 1:800), MYL2 (Protein Tech Group 10906-1-AP, 1:200), MYH11 (Dako, M0851, diluted 1:1000) and GJA1 (Abcam ab11370, 1:1000). Secondary antibody staining was performed for 1 h at room temperature using anti-mouse and anti-rabbit Alexa Fluor 568 and 647 conjugated antibodies (all ThermoFisher). Following staining, plates were incubated with 1 µg/ml DAPI PBS for 1 min and stored at 4 °C in PBS.

**Quantitative PCR.** Analysis of gene expression by quantitative PCR was performed, as described previously[16,64]. Expression levels of transcripts were normalized to the averaged expression of the housekeeping genes GAPDH and SRP72. Taqman probes were used for all genes (ThermoFisher).

**Calcium imaging.** Differentiated cells (Day 10) were seeded onto Geltrex coated optical tissue culture treated 96 well plates at 1.5 × 10$^4$ cells/cm$^2$. Cells were analyzed 4-6 days post plating. Cells were loaded with Fluo-4-AM (5 µM, Molecular

Probes) 30 min prior to analysis. Intracellular calcium concentration ([Ca$^{2+}$]$_i$) was measured by illuminating myocytes (at ×10 magnification) once per second with light (488 nm) and emission recorded using the GFP filter set of a Nikon A1R confocal microscope (Japan)[65]. Cells displaying oscillating fluorescence were considered to be spontaneously active and changes in fluorescence intensity were measured for 10 min to determine changes in [Ca$^{2+}$]$_i$.

For analysis of electrical conduction through cardiomyocyte cultures, cells were seeded onto Geltrex coated 24 well plates at 1.25 × 10$^5$ cells/cm$^2$ and were analyzed using the calcium imaging method described. Images were collected at 8 frames per second at ×4 magnification. Two regions (~500 µM$^2$) separated by 1.3 mm were selected and changes in [Ca$^{2+}$]$_i$ measured over a 10 s period. Background fluorescence was subtracted and changes in fluorescence intensity in the two regions were plotted against each other (example shown in Fig. 2a). Regression values were plotted for each pair of regions (GraphPad Prism v6) and the mean ± SEM of these values calculated was used to determine the correlation between the two regions as a surrogate measure of conduction efficiency.

**Multi-electrode array.** Differentiated cells were harvested using TrypLE Select and aggregated by centrifuging cells (4 min at rcf 478) suspended in basal differentiation media at 1.0 × 10$^4$ cells per well in low adherence U bottom 96 well plates. At 24 h post aggregation, aggregates were seeded onto Geltrex coated 6 well microelectrode arrays (Multi Channel Systems). At 24-48 h post seeding, basal differentiation media was exchanged and recordings made following equilibration. Adrenergic responses were analyzed with isoproterenol hydrochloride (Sigma, I6504) dissolved in H$_2$O. Data was recorded and analyzed using MC Rack software (Multi Channel Systems). Field potential duration measurements were corrected using Fridericia's repolarisation correction formula (QTcF).

**Whole-cell patch clamp.** Differentiated cells were seeded onto Geltrex coated glass bottom 35 mm culture dishes as single cells (World Precision Instruments). Spontaneous action potentials (APs) were recorded from 4-6 days post plating using a HEKA EPC10 Double patch clamp amplifier at room temperature (HEKA Elektronik, Germany). Borosilicate pipettes (Harvard Instruments) with an input resistance from 1-3.5MΩ were filled with 117 mM KCl, 10 mM NaCl, 2 mM MgCl$_2$, 1 mM CaCl$_2$, 11 mM EGTA, 2 mM Na-ATP, and 11 mM HEPES. The pH was adjusted to pH 7.2 with KOH. Cells were bathed in a solution containing 135 mM NaCl, 5 mM KCl, 5 mM HEPES, 10 mM glucose, 1.2 mM MgCl$_2$, and 1.25 mM CaCl$_2$. The pH was adjusted to 7.4 with NaOH.

Cells were patch clamped in whole-cell voltage-clamp mode. Slow and fast-capacitance were compensated for using Patchmaster data acquisition software (HEKA) and signals were filtered with a 10 kHz low-pass Bessel filter. The amplifier was then switched to current-clamp mode to measure the voltage wave-form. Spontaneous action potential firing was recorded without current injection. Data analysis was performed using custom scripts written with MATLAB (Mathworks) (script provided in Supplementary Methods).

**Cardiac organoids.** Cardiac organoid formation and growth was adapted from ref. [25]. Briefly, initial cardiac differentiation was induced in monolayers using RPMI-B27 medium containing 5 ng/mL BMP-4 (RnD Systems), 9 ng/mL Activin A (RnD Systems), 5 ng/mL FGF-2 (RnD Systems), and 1 µM CHIR99021 (Stem Cell Technologies) with daily medium exchange for 3 days. Subsequently, cultures were maintained in RPMI-B27 supplemented with 5 µM IWP-4 (Stem Cell Technologies) for 3 days to guide specification into cardiomyocyte and stromal cell lineages. Cultures were maintained in RPMI-B27 with medium exchange every 2 days for a further 9 days. On Day 15 single cell suspensions were generated by digestion in collagenase type I (Sigma) in 20% Foetal Bovine Serum in phosphate buffered saline for 60 min at 37 °C followed by 0.25% trypsin-EDTA for 10 min and filtration through a 100-µm mesh cell strainer (BD Biosciences). For cardiac organoid formation 5 × 10$^4$ day 15 cells in CTRL media (α-MEM GlutaMAX, 10% Foetal Bovine Serum, 200 µM L-ascorbic acid 2 phosphate sesquimagnesium salt hydrate, and 1% Penicillin/Streptomycin) were mixed with Matrigel (9%) and collagen I (2.6 mg/ml; Devro) in a total volume of 3.5 µl. Subsequently, the cell/Matrigel/collagen I mixture was added to Heart-Dyno constructs (below) and centrifuged. The Heart-Dyno was then centrifuged at 100 × g for 10 s to ensure the hCO form halfway up the posts. The mixture was then gelled at 37 °C for 30 min prior to the addition of CTRL medium to cover the tissues (150 µl/hCO). The Heart-Dyno design facilitates the self-formation of tissues around in-built PDMS exercise poles (designed to deform •0.07 µm/µN). The medium was changed every 2–3 days (150 µl/hCO).

Heart-Dyno's constructs were manufactured using SU-8 photolithography and PDMS molding[25]. Briefly, microfabricated cantilever array designs were drafted with DraftSight (Dassault Systems) and photomasks of the design were then plotted with an MIVA photoplotter onto 7-inch HY2 glass plates (Konica Minolta) followed by SU-8 photolithography on 6-inch silicon wafer substrates (•700 µm). Silicon wafers were cleaned and degassed at 150 °C for 30 min. Subsequently, SU-8 2150 photoresist (Microchem) was spin coated to build the SU-8 to the required thickness and the final wafer exposed to UV (1,082 mJ/cm$^2$). The Heart-Dyno was molded by soft lithography with PDMS (Sylgard 184; Dow Corning; mixed in 10:1 ratio of monomer:catalyst), with curing at 65 °C for 35 min. The molds were placed

into 96-well plates, sterilized with 70% ethanol and UV light, washed with PBS, and coated with 3% BSA (Sigma, A2153).

**Transmission electron microscopy**. Cardiomyocytes were sorted by flow cytometry based on the expression of eGFP and VCAM1 at day 10 of differentiation and replated onto Nunc Thermanox 13 mm coverslips (Thermo Scientific 174950) coasted in Geltrex (Invitrogen A1413202). The cardiomyocytes were cultured until day 21 in RPMI1640 with B27 supplement and then fixed in a 1.5% Glutaraldehyde, 1.5% Paraformaldehyde mix in PBS. Transmission electron microscopy sample processing and imaging was performed as a fee for service at the Bio21 advance microscopy facility. A minimum of 20 images per cell line were captured on a Tecnai F30 TEM instrument. Image analysis was perform blinded to genotype.

**RNAseq**. GFP positive cells were FACS sorted on day 10 of differentiation and snap frozen. Cells were subsequently thawed and collected in PBS to generate 3 pools of $>1.0 \times 10^6$ cells for both $NKX2\text{-}5^{eGFP/w}$ and $NKX2\text{-}5^{-/-}$ cell lines. RNA was extracted using a High Pure RNA Isolation kit (Roche) and 1 μg was analyzed for RNA integrity and submitted for sequencing using the Illumina platform (Australian Genomic Research Facility). Tophat2 aligner was used to map the 100 bp single end reads to the human reference genome (hg19). The uniquely mapped reads were summarized across genes with featureCounts (Rsubread v1.20.6)[66] using RefSeq gene annotation (hg19). Lowly expressed genes were filtered out (less than one count per million in fewer than three samples), leaving 14,458 genes for further analysis. The data was TMM normalized, voom transformed[67], and differential expression assessed using empirical Bayes moderated $t$-tests from the R Bioconductor limma package[68]. Design matrix included factor for day of cell sorting. Differential gene expression was determined by fold change <2 and with adjusted $p$ values > 0.1 (moderated $t$-test). Data have been deposited on the GEO database under accession code GSE89443 (https://www.ncbi.nlm.nih.gov/geo/query/acc.cgi?acc=GSE89443).

The IUPHAR/BPS Guide to PHARMACOLOGY database (http://www.guidetopharmacology.org) was used to identify ion channel and transporter genes[69]. Myofibrillar components and smooth muscle associated gene lists were taken from[70]. GO analysis was performed using ToppGene suite (https://toppgene.cchmc.org/).

**ChIPseq**. Differentiated cells were fixed on day 10 for ChIPseq studies. Cells were PBS washed and fixed for 10 min at room temperature with shaking in fresh methanol free formaldehyde (ThermoFisher) diluted to 1% in cold PBS. Formaldehyde was quenched with glycine, cells PBS washed and snap frozen. For ChIP, protease inhibitor cocktail was used in all buffers (Roche). Aliquots of ~5 × $10^6$ cells were re-suspended in lysis buffer (1% SDS, 10 mM EDTA, 2 M Tris–HCl) and incubated on ice for 20 min. Cells were sonicated using a focused ultrasonicator (Covaris) using peak voltage 80 W and duty cycle 3% for time 25 min. Sonicated samples were diluted in dilution buffer (0.01% SDS, 1.1% Triton X-100, 1.2 mM EDTA, 16.7 mM Tris-HCl, 165 mM NaCl) and pre cleared for 4 h at 4 °C with blocked Protein A conjugated magnetic beads (ThermoFisher). Beads were removed and supernatant incubated overnight at 4 °C in dilution buffer with 5 μg of one of two NKX2-5 antibodies validated by immunohistochemistry (Abcam ab35842, Santa Cruz sc-14033) or IgG as a control (Sigma). Blocked beads were added to the supernatants, and incubated for 4 h at 4 °C. Beads were washed in dilution buffer, low salt wash buffer (0.5% sodium deoxycholate, 0.1% SDS, 1% NP-40, 1 mM EDTA, 50 mM Tris-HCl, 150 mM NaCl), high salt wash buffer (as low salt, 500 mM NaCl) and TE. Bound protein:DNA complexes were eluted (1% SDS, 100 mM NaHCO₃) and cross linking reversed by incubating overnight at 65 °C in a final concentration of 0.3 M NaCl. DNA was purified using a PCR purification kit (QIAGEN). Samples were quantified, and if required, pooled and vacuum concentrated. Sequencing was performed using Illumina chemistry at the Australian Genomic Research Facility. All sequenced files were trimmed for adapters using program trimmomatic. Bowtie2 (v2.10) using default parameters was used to map the 100 bp single end reads to the human reference genome (hg19). The Macs1.4 (version 1.4.2) program was used to call ChIP peaks from three independent samples (two prepared using Abcam antidbody, one using Santa Cruz antibody) using an input sample (IgG antibody) as a control. To ensure only high quality reproducible peaks were used for further analysis, only the peaks (or part of the peaks) that were detected by Macs1.4 (individual peaks $p$ values < 1e-5) in all three samples were used for downstream analysis. Meme chip (http://meme.nbcr.net/meme/cgi-bin/meme-chip.cgi) was used to identify potential motifs within the called chip peaks. The Genomic Regions Enrichment of Annotation Tool (GREAT), version3.0.0, was used to predict the function of cis-regulatory regions. Parameters used; Human hg19 genome, whole genome as background. Default gene –region association rules were used. Data have been deposited in the GEO database under accession code GSE89457 (https://www.ncbi.nlm.nih.gov/geo/query/acc.cgi?acc=GSE89457).

**GAPTrap rescue experiments**. We utilized the GAPTrap strategy to engineer cell lines expressing transcription factors of interest from the GAPDH locus, as described previously[54]. GAPTrap targeting vectors were modified such that sequences encoding NKX2-5, HEY2 and IRX4, all fused in-frame to the ligand-binding domain of the human Estrogen Receptor[50], were placed immediately 3′

of the T2A peptide cleavage signal (Supplementary Fig. 4). All cloning was performed using the InFusion HD cloning system (TaKaRa, 638910). To induce nuclear translocation of the NKX2-5::ER, HEY2::ER and IRX4::ER fusion proteins the ligand 4-Hydroxytamoxifen (Sigma, T176) was added to cell culture media at final concentration of 0.5 μM. For rescue experiments, 4-Hydroxytamoxifen (4-OHT) was added from differentiation day 5 onwards.

**Western blots**. Proteins were extracted from differentiated hESCs-cardiomyocytes (day 14) by incubating cultures in ice cold RIPA buffer supplemented with proteinase inhibitors (Roche) for 10 min and scrapping the cells and collecting in Eppendorf tubes. The insoluble fraction was removed by centrifugation and protein extracts were snap frozen in liquid nitrogen. 50 μg of whole cell protein extract was separated on NuPAGE Novex 4–12% Bis-Tris Midi Protein Gels (ThermoFisher). Proteins were transferred to Amersham Hybond-P PVDF Membrane according manufactures protocols (RPN2020F, GE Healthcare, Life Sciences). Membranes were blocked in 5% BSA in Tris-Buffered Saline with 0.01% Tween 20 (TBS-T). To detect antigens the membranes were incubated for 1 h in primary antibodies (NKX2-5 (Santa Cruz sc-14033 or Abcam ab35842, diluted 1:100; GAPDH Cell Signalling, 51745 and GJA1 (Abcam ab11370, 1:100)) in 1% BSA in TBS-T, then washed 3 times in TBS-T followed by a 1 h incubation with horse radish peroxidase conjugated secondary antibody (Jackson Immuno Research labs., 115-035-003, 1:1000) an then washed 3 times in TBS-T. Proteins were detected with the Amersham ECL Western blotting detection kit according to manufactures protocol (GE Healthcare, RPN2108). For fluorescent based Westerns antibodies were detected with an appropriate secondary antibody conjugated to Alexa-647 (goat-anti mouse A-21235; goat-anti rabbit A-21245, ThermoFisher) diluted 1:1000 in TBS-T and imaged on a ImageQuant LAS 500 (GE Healthcare).

All uncropped Western blots can be found in Supplementary Fig. 6.

**Luciferase reporter assays**. The NKX2-5 expression vector was purchased from GenScript (pcDNA3.1+/C-(K)-DYK-NKX2-5, OHu19766D) and expression clones for Tbx20 and Gata4 have been previously described[49]. Luciferase reporters were generated by placing the 5′ (DNA sequence—chr6:126,140,479-126,141,036) and 3′ (DNA sequence—chr6: 126,604,942-126,605,309) HEY2 putative enhancer elements in the pGL3-Promoter vector (Promega, USA, E1761). For transfection assays HEK 293 T cells were plated at $0.5 \times 10^5$ cells per well in 48 well plates and placed in a 10% $CO_2$ incubator at 37 °C. After 24 h the cells were transfected using the Viromer Yellow transfection reagent (Lipocalix, Germany, VY-01LB-00) mixed with the plasmids in the following amounts: 50 ng Enhancer reporter, 5 ng Renilla reporter (pRL-CMV), 50 ng each of Gata4 and Tbx20 expression vectors and 200 ng of pcDNA3.1+/C-(K)-DYK-NKX2-5. Transcriptional activity was determined using Dual Luciferase Reporter Assay kit (Promega, USA, E1910) and controlled for transfection efficiency by determining Renilla luciferase activity. Transfections were performed in triplicate and statistical analysis performed using one-way Anova (GraphPad Prism Software). Data are presented as fold activity relative to the corresponding reporter when co-transfected with empty expression plasmids.

**Data availability**. The authors declare that all data supporting the findings of this study are available within the article and its Supplementary Information files or the GEO database https://www.ncbi.nlm.nih.gov/geo/query/acc.cgi?acc=GSE89443, https://www.ncbi.nlm.nih.gov/geo/query/acc.cgi?acc=GSE89457 or from the corresponding author (D.A.E.) upon reasonable request.

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

## Acknowledgements

Supported by the Victorian Government's Operational Infrastructure Support Program and Australian Government National Health and Medical Research Council (NHMRC) Independent Research Institute Infrastructure Support Scheme (IRIISS). D.J.A. supported by the European Union's Seventh Framework Programme (FP7/2007-2013) under grant agreement PIOF-GA-2010-276186. R.P.H. supported by grants from the NHMRC (1074386; 573732) and the Australian Research Council Strategic Initiative in Stem Cell Science (SR110001002). R.P.H. held an NHMRC Australia Fellowship (573705). A.G.E. and E.G.S. are Senior Research Fellows of the NHMRC. Qatar National Research Fund (NPRP 09-1087-3-274) supported D.A.E., A.R., E.G.S. Research in the laboratories of D.A.E., A.G.E. and E.G.S. was supported by the NHMRC, the Australian Research Council Strategic Initiative in Stem Cell Science (SR110001002) and the Stafford Fox Medical Research Foundation. The Royal Children's Hospital Foundation and Paceline provided support for D.A.E. and M.M.C.

## Author contributions

D.J.A. and D.A.E. conceived the study, performed experiments, collected and interpreted data and wrote the manuscript. D.I.K., K.K., J.M.H., R.J.M., D.G.P., E.L.Q., A.R.L., D.A., T.L., E.S.N., R.P.D., S.C., R.P., J.E.H., E.R.P., M.W.C., C.L.C., performed experiments and collected and interpreted data. K.M.B. conducted bioinformatic analysis. A.R., L.M.D., A.O., M.M.C., S.P., designed experiments and provided data analysis and interpretation. R.P.H., C.L.M., A.G.E., E.G.S. designed experiments and contributed to data analysis, and contributed to editing the manuscript.

## Additional information

**Competing interests:** The authors declare no competing interests.

