## [Peer Review File(PDF 960 kb) · Nature Communications]

Reviewers' Comments:

Reviewer #1 (Remarks to the Author)

The study by Anderson et al examines Nkx2.5-dependent changes in cardiomyocyte lineage maturation using gene targeted human embryonic stem cells. Nkx2.5 null hESC were generated and downstream effects on cardiomyocyte differentiation, function, and gene expression were analyzed. As is observed in mice, loss of Nkx2.5 does not prevent differentiation and beating of cardiac progenitors. However, expression of cardiac maturation, gap junctions and ion channels was reduced, as would be predicted from previous studies. Altered expression of VCAM1 and PDGFRalpha, previously identified as cardiac lineage markers by this group, also was observed. The main novel finding is a link between Nkx2.5 and Hey2 as evident in a partial rescue of the Nkx2.5 null cells with Hey2 overexpression.

Comments

1. Most of the findings and downstream targets identified in the current study are confirmatory of what has been found for mouse nkx2.5.

2. The main novel finding is the identification of Hey2 as a mediator of Nkx2.5-dependent maturation of cardiomyocytes. This conclusion is based on qPCR analyses of target genes in the cultured cardiomyocytes with Hey2 induction. Additional data on restoration of cardiomyocyte contractile apparatus, gap junctions, and function by Hey2 overexpression in Nkx2.5-null hESC should be provided in support of the proposed rescue mechanism.

3. The Nkx2.5null cardiomyocyte phenotype is not well characterized. It would be good to see the sarcomeric architecture and high magnification images of the gap junctions as indicative of the maturation state of the cells.

4. Expression of genes associated with smooth muscle, second heart field and AV canal all could be indicative of lack of maturation of Nkx2.5-null cardiomyocytes after differentiation, since these genes are expressed in immature populations of cardiomyocytes. This would be consistent with the mouse null phenotype.

5. It is not clear what the implications of altered expression of VCAM1 and PDGFRa are in terms of cardiac maturation in vivo. Neither of these genes is cardiac-specific and neither has been implication in heart development in vivo that I know of. In addition, direct regulation of PDGFRa gene expression by Nkx2.5 has not really been disproven as is concluded by the ChIP with a limited region of the locus, since regulatory elements can be far removed from the linear location of the transcriptional start site.

Reviewer #2 (Remarks to the Author)

This manuscript by Anderson et al. examines the mechanism of how Nkx2-5 regulates cardiogenesis. The authors generate an Nkx2-5 null and heterozygous human ESC and use these lines to examine the defects that ensue due to Nkx2-5 deletion. The role of Nkx2-5 has been extensively studied during murine development but less so in the cellular differentiation of human cardiomyocyte. In that regard the ESC system provides an attractive opportunity. Although on the whole this is a well thought out and well performed study, addressing the concerns outlined below is required.

Major concerns:

1. All of the conclusions of the manuscript are drawn from two isogenic cell lines with a heterozygous or homozygous insertion of the eGFP in the Nkx2-5 locus. Although the parent cell

line is well validated, the fact that no independent line of evidence is provided to validate the findings is an important weakness. This is of further concern given *in vivo* evidence Nkx2-5 may exhibit haploinsufficiency. It would be useful to validate the major findings of this report with another completely independent cell line. This can now be accomplished relatively easily given the high efficiency of differentiation of multiple ESC and hiPSC lines and the relative ease of generation of biallelic deletions using current genome engineering technologies.

2. Likewise, consideration should be given to bolstering the developmental arguments of this interaction using mouse embryonic studies.

3. The authors suggest that although the Hey2 promoter has multiple Nkx2-5 binding site, Nkx2-5 may indirectly activate Hey 2 expression (e.g. via Coup-TF or BMP signaling). Determining whether the interaction between the Hey2 promoter and Nkx2-5 is directly responsible for the activation of Hey 2 promoter would be of interest and would significantly improve the manuscript. This may be done with suppression expressions of the Coup-TF or BMP signaling pathway in the PSC used in the study or alternatively using cell lines that do not express these signaling pathways and determining whether Nkx2-5 is either necessary or sufficient for the expression of Hey2 using a transcriptional readout of that promoter.

Additional concerns:

1. In Figure 1, eGFP expression appears to be nuclear. Are the investigators using a nuclear GFP and not eGFP? If so, this should be made explicit. Otherwise an explanation regarding the nuclear localization of eGFP should be provided.

2. Figure 1D: Is the qPCR performed in FACS sorted eGFP⁺ cells? This data seems to suggest that eGFP⁺ cells are very heterogeneous. It would seem that this heterogeneity may undermine some of the conclusions that the authors are drawing and this should be investigated—are the eGFP low cells at different developmental/differentiation stage than the high population, is Nkx2-5 expression (at least in the het cell line) correspondingly lower? Which population is driving the observations from this study?

3. Figure 1G: It appears that there are two different eGFP⁺ populations, an eGFP high and low population. Can the authors provide any explanations of these findings and/or an analysis of their significance?

4. Figure 2A: The Nkx2-5^{-/-} Fluo4-AM tracing appears to be very noisy with little discernable signal. While this may be due to inherent defects in calcium handling in these cell types, the poor signal to noise ratio in these cells makes it difficult to assess the coordination of calcium flux in adjacent cells. Perhaps the ^{-/-} cells are simply more sensitive to the known cytotoxic effects of calcium dyes.

5. The contractility analysis of the organoids is superficial. The investigators need to define whether the apparent defect in contractility is due to abnormal electromechanical coupling (as the data would seem to suggest) or to an actual defect in cardiomyocyte contractile function--It is possible that the functional difference between the two cell types is only due to poor electrical coupling of the organoids and not to a contractility defect. This needs to be sorted out potentially through single cell analysis of contractility (multiple options from multiple groups available for this type of analysis).

6. Figure S1. D-too many cells are off the scale

7. Figure S1E- cellular constructs not clear, example seems to be of an empty well.

Reviewer #3 (Remarks to the Author)

Review for "NKX2-5 regulates human cardiomyogenesis via a HEY2-dependent transcriptional network" (Anderson et al).

To determine a molecular mechanism by which the cardiac transcription factor NKX2-5 functions in developing cardiomyocytes, Anderson et al depleted the wild type allele of NKX2-5 in a hESC model in which one copy of NKX2-5 was replaced with GFP. NKX2-5^{-/-} cells differentiate into cardiomyocytes but have delayed onset of differentiation. NKX2-5^{-/-} iCMs also retain PDGFRa

expression and have reduced VCAM1 expression, suggesting perturbed differentiation.

The authors then performed contractility analyses to demonstrate impaired electrophysiology in NKX2.5^{-/-} cells. These cells demonstrate abnormal contraction as shown by calcium oscillation calculations, MEA analysis, and patch clamp analysis. They then generated bioengineered cardiac organoids from these cells to determine that mutations in NKX2-5 exhibited decreased contractile force.

To determine the NKX2.5 gene network the authors performed RNA-seq analysis with ChIP-seq analysis to determine direct targets of NKX2.5. RNA seq analysis determined that ventricular specific genes were reduced in the NKX2.5 knockout cells while cardiac progenitor genes were upregulated in the knockout cells. ChIP-seq analysis showed that NKX2.5 was bound near genes involved in cardiac development. When ChIP peak data was overlaid with RNA-seq expression data it was found that genes that are directly bound and activated by NKX2.5 are involved in channel activities, and genes that are directly bound and repressed by NKX2.5 are involved in calcium binding and neurogenesis. The authors then examined gap junction proteins to explain differences in ion channel expression and found that gap junction GJA1 was not localized to the gap junctions. They also noted aberrant expression of genes required for the SHF, AVC, and OFT.

The authors then determined the transcription factor network that interacts with NKX2.5 and found a number of transcription factors (including HEY-2 and the IRX cluster) that are misregulated in NKX2.5 null iCMs. They then performed a rescue assay using GAPTrap and found that temporal introduction of NKX2.5 and HEY2, but not IRX4, rescued the null phenotype in terms of VCAM1 expression, suggesting that HEY2 is one of the critical mediators of the NKX2.5-dependent transcriptional network.

Major issues

- 1) Figure 1: a) The hESCs seem to differentiate fairly well in culture, with >80% of the cells expressing GFP and ACTN2 by day 10 of differentiation. In supplementary Figure 1H, cell sorting analysis was performed to characterize ACTN2 and GFP expression in culture at Day 14. Although many of the cells (please provide quantitation here) are ACTN2⁺/GFP⁺, and some cells that are ACTN2⁻/GFP⁻, there are also a significant number of cells that are ACTN2⁺/GFP⁻. Please comment on why that might be and how these cells may affect your analysis.
- 2) All comparisons in the manuscript are between the heterozygous and homozygous null cell. Congenital heart disease is caused by haploinsufficiency of NKX2-5. Why are there no comparisons of heterozygous mutant cells to wild-type HES3 cells? For example, how many NKX2-5 binding sites differ between wildtype and heterozygous cells? How many differences in RNA expression?
- 3) Fig 3. Differences in RNA expression between heterozygous and homozygous NKX2-5 cells and comparison to NKX2-5 CHIP-Seq peaks. Again, this data should also be compared to wildtype cells. Do heterozygous cells have the same NKX2-5 binding peaks as wildtype cells?
- 4) RNA-seq analysis for GFP⁺ enriched cardiomyocytes at Day 10 of differentiation: PDGFRa is shown to be enriched in NKX2.5 null cells, but earlier in the text PDGFRa is reported as unchanged up to day 14. Please clarify.
- 5) Materials and methods paragraph for the GAPTrap assay seems to be missing.
- 6) NKX2.5, HEY2, and IRX4 are induced by adding 4-OHT at day 4. Why does the VCAM-APC expression increase so drastically before induction of these proteins for the rescue experiment in supplemental figure 4h?
- 7) In the discussion NKX2.5-HEY2 regulatory relationship is postulated to be indirect. This should be tested by coimmunoprecipitation experiments in the iCMs. Additionally, transcriptional assays should be performed to test this hypothesis. Are HEY2 binding motifs upregulated near NKX2.5 binding sites?

Minor issues

Supplementary Figure 1C: The karyotype image is too small to actually determine if any defects in karyotype are present.

Supplementary Figure 1: For all flow cytometry analyses, please provide quantitation for your data.

Figure 2A-B: Please specify the time point for the cells used in this analysis.

Why did you perform the ChIP analyses using cells at Day 10 when all other analyses were performed at Day 14?

Figure 3i. It is customary to place a black line on a western blot designating a break in the image. Please also include the molecular weights or provide an unaltered image. There are a number of typos in the results and the materials and methods section. Please correct typos accordingly.

Page 16. Please provide catalog information for PE/Cy7 conjugated secondary antibody.

There are many instances in which the figures that are referenced in the results section do not correspond with the text. For example, on page 5, for your MEA analysis, the basal rate of contraction is shown in Figures 2d and 2e, and the prolonged field potential is shown in Figures 2d and 2f. Please fix the references to your figures in the second paragraph under the "Perturbed electrophysiology..." header. Please correct the references to figures throughout the manuscript. In particular Figure 3F-G, ion channel and transporter genes had altered expression in NKX2-5 cells, and that differential expression was maintained throughout differentiation. However, none of the genes highlighted in figure 3F are reported in 3G, and vice versa. In addition to KCNH2a/2b, please show the RNA-seq expression analysis in 3G with the genes highlighted in 3F (SLC47A1, GRIA1, etc). The same is true for transcription factor expression (Figure 4b). GATA4 and TBX5 expression is not dependent on NKX2.5 and reference figure 4B, but this statement is not supported by the data presented in Figure 4b.

Response to Reviewers

Summary.

We thank the reviewers for their detailed and considered comments on our manuscript. We have addressed the reviewer's concerns (in blue text below) and added new data (red text below and within the manuscript) with the major new findings summarized below:

- The *NKX2-5* null phenotype reported has been confirmed by findings using (in?) a second hESC background, the widely used H9 cell line. Cardiomyocytes from H9 *NKX2-5*^{-/-} cells have reduced VCAM1 levels and have a gene expression profile that is consistent with that observed in the HES3 *NKX2-5* null cardiomyocytes (Supplementary Fig. 1k, l; Supplementary Fig. 3a).
- We demonstrate by transmission electron microscopic analysis that the cytoskeletal architecture of *NKX2-5* null cardiomyocytes is perturbed. This new data is presented in Fig. 2i and Supplementary Fig. 2g.
- We show that, at the protein level, there is no discernible evidence for *NKX2-5* haploinsufficiency in our system (Supplementary Fig. 1f). At a transcriptional level there is no detectable difference of expression of the key markers IRX4, HEY2 and VCAM1 between wildtype and *NKX2-5*^{eGFP/w} cultures (Supplementary Fig. 3b). Furthermore, protein levels of GJA1 (connexin 43) are similar between wildtype and *NKX2-5*^{eGFP/w} cardiomyocytes (Fig. 4i). These data are consistent with our previous work, clinical evidence and studies in the mouse that suggest heterozygosity for *NKX2-5* has very minor consequences that are unlikely to impact on our findings (see below).
- We demonstrate that *NKX2-5* is able to activate transcription of *HEY2* from nearby putative enhancers in a transient transcriptional assay in HEK 293T cells (Supplementary Fig. 4f).
- Western blots for GJA1 provide further support to the genetic rescue experiments that demonstrate the restoration of gap junctions (connexin 43) (Fig. 4i).

We note that all three reviewers comment on the novelty of the *NKX2-5*/*HEY2* regulatory axis identified in this work. We concur that this transcriptional mechanism of cardiomyocyte maturation is of importance in heart development and function. We believe that our new data and incorporation of the changes suggested by the reviewers substantially improve the manuscript.

Reviewer #1 (Remarks to the Author):

The study by Anderson et al examines *Nkx2.5*-dependent changes in cardiomyocyte lineage maturation using gene targeted human embryonic stem cells. *Nkx2.5* null hESC were generated and downstream effects on cardiomyocyte differentiation, function, and gene expression were analyzed. As is observed in mice, loss of *Nkx2.5* does not prevent differentiation and beating of cardiac progenitors. However, expression of cardiac maturation, gap junctions and ion channels was reduced, as would be predicted from previous studies. Altered expression of VCAM1 and PDGFRalpha, previously identified as cardiac lineage markers by this group, also was observed.

The main novel finding is a link between *Nkx2.5* and *Hey2* as evident in a partial rescue of the *Nkx2.5* null cells with *Hey2* overexpression.

Comments

1. Most of the findings and downstream targets identified in the current study are confirmatory of what has been found for mouse *nkx2.5*.

Whilst overlap between *NKX2-5* activity in the mouse and human is to be expected given that *NKX2-5* is a highly conserved member of the cardiac gene regulatory network, our data clearly indicate that *NKX2-5* has been deployed in human specific gene programs. Primarily, we identify a new *HEY2* dependant transcriptional axis that has never been reported in mice. Importantly, recent highly detailed molecular dissection of murine *Nkx2-5* function did not identify *Hey2* as an integral component of the *NKX2-5* transcriptional network^{1,2}. This demonstrates a key strength of human pluripotent stem cells as a model to study human heart development.

2. The main novel finding is the identification of *Hey2* as a mediator of *Nkx2.5*-dependent maturation of cardiomyocytes. This conclusion is based on qPCR analyses of target genes in the cultured cardiomyocytes with *Hey2* induction. Additional data on restoration of cardiomyocyte contractile apparatus, gap junctions, and function by *Hey2* overexpression in *Nkx2.5*-null hESC should be provided in support of the proposed rescue mechanism.

To provide additional support for the role of *HEY2* as a mediator of *NKX2-5* activity, we determined if *GJA1* (connexin 43) levels are restored in *NKX2-5* knockouts expressing *HEY2::ER* from the *GAPDH* locus. Importantly, mRNA for *GJA1* is not differentially expressed between the *NKX2-*

$5^{eGFP/w}$ and $NKX2-5^{-/-}$ cardiomyocytes. If GJA1 is not correctly incorporated into gap junctions the protein is rapidly degraded³ and, therefore, GJA1 protein levels are informative with respect to gap junction and intercalated disc formation in our culture system. Western blot data demonstrates that GJA1 levels are rescued by HEY2 expressed via the GAPTrap system (Fig 4i). Thus, these data support a regulatory role for HEY2 in establishing the cardiomyogenic gene program.

We also highlight that, in addition to Q-PCR analysis, our initial manuscript illustrated that HEY2 functionally compensates for NKX2-5 as shown by i) increased VCAM1 expression levels (Fig. 4f) and ii) the restoration of contractile synchronicity to cardiac monolayers (Fig. 4h). Importantly, all experiments assessing synchronicity were performed blind to genotype and, thus, these data were collected in an unbiased manner. Taken together, these biochemical, gene expression and functional data demonstrate HEY2 is an important transcriptional component of the NKX2-5-dependant gene regulatory network.

The new data is presented in the text as follows:

...and GJA1 protein levels were restored in HEY2::ER rescued cultures to levels comparable to both $NKX2-5^{eGFP/w}$ and wildtype cardiomyocytes (Fig. 4i).

and

“Conversely, induced expression of HEY2 partially rescued the NKX2-5 phenotype, including restoring GJA1 levels, without up regulating IRX4.”

Figure 4i is shown below:

i

(i) Western blot showing restoration of GJA1 (connexin 43) levels by HEY2 and that wildtype (HES3) and $NKX2-5^{eGFP/w}$ GJA1 levels are comparable.

3. The $Nkx2.5$ null cardiomyocyte phenotype is not well characterized. It would be good to see the sarcomeric architecture and high magnification images of the gap junctions as indicative of the maturation state of the cells.

To address this concern, we have provided additional transmission electron micrograph images of both $NKX2-5^{eGFP/w}$ and $NKX2-5$ null cardiomyocytes (Fig. 2i and Supplementary Fig. 2g). These images demonstrate that the sarcomeric architecture is disorganised in the $NKX2-5$ null lines. Furthermore, these TEM data support the confocal immunocytochemistry data presented in Figures 3 h, k, l, which show perturbed expression of MYL2 (myosin light chain 2), MYH11 (smooth muscle myosin heavy chain) and GJA1 (connexin 43). Taken together, this data set provides a more detailed profile of the $NKX2-5$ null cardiomyocytes. While these TEM images did not identify gap junctions, we continue to investigate the role of $NKX2-5$ in the regulation of gap junction formation. In this context, as described in response to the Reviewer's point 2 (above), we show that HEY2 is able to rescue GJA1 protein levels, suggesting that GJA1 turnover is reduced

because the protein has been stably integrated into gap junctions (Fig. 4i). Thus, these data provide support for the notion that NKX2-5 drives the regulation of cardiomyocyte maturation and formation of the intercalated discs.

These data are described in the text as follows:

Consistent with impaired muscle function the sarcomeres of *NKX2-5* null cardiomyocytes are disorganized (Fig. 2 i and Supplementary Fig. 2g).

and

...and GJA1 protein levels were restored in HEY2::ER rescued cultures to levels comparable to both *NKX2-5*^{eGFP/w} and wildtype cardiomyocytes (Fig. 4i).

The data from Figure 2i and Supplementary Figure 2g are shown below:

(i) Transmission electron micrographs show that *NKX2-5* null cardiomyocytes have disorganised sarcomeres compared to *NKX2-5*^{eGFP/w} cardiomyocytes (see also Supplementary Fig. 2g). Scale bar = 1 μ M.

(g) Transmission electron micrographs show that *NKX2-5* null cardiomyocytes have disorganised sarcomeres compared to *NKX2-5*^{eGFP/w} cardiomyocytes. Dashed boxes are areas shown figure 2. Scale bar = 1 μ M.

4. Expression of genes associated with smooth muscle, second heart field and AV canal all could be indicative of lack of maturation of *Nkx2.5*-null cardiomyocytes after differentiation, since these genes are expressed in immature populations of cardiomyocytes. This would be consistent with the mouse null phenotype.

We concur with this statement as we indicated in our manuscript:

“Collectively, these data provide molecular evidence supporting the hypothesis that NKX2-5 is required for the progression of cardiomyocytes into specialised ventricular phenotype, ...”

and,

“Together, these data suggest that progression to a ventricular cardiac phenotype is blocked in NKX2-5^{-/-} cardiomyocytes and NKX2-5 is required to repress the ancestral smooth muscle genetic program. Alternatively, or additionally, it is also possible that in the absence of NKX2-5, heart progenitor cells with cardiomyocyte and smooth muscle potential^{26, 37} may preferentially adopt a smooth muscle fate.”

and,

“This data shows that both important developmental genes and markers of specialized non-myocyte lineages are dysregulated in NKX2-5 null cells and the presence of NKX2-5 at these loci supports an important, conserved role for human NKX2-5 in these developmental processes and cell types^{8, 38}.

and,

“Furthermore, gene expression and genomic binding profiling demonstrated dysregulation of the ventricular myogenic program and key progenitor genes, with higher expression of smooth muscle, second heart field and atrioventricular genes and loss of normal ion channel gene expression in NKX2-5^{-/-} derived cardiomyocytes. These changes subsequently manifest as reduced contractile force and asynchronous contraction of cardiac sheets in the NKX2-5 mutant cells.”

Thus, whilst highlighting the immature nature of the NKX2-5 null cardiomyocytes we have also placed our data within the context of established cardiac developmental paradigms, in line with the reviewer’s suggestions.

5. It is not clear what the implications of altered expression of VCAM1 and PDGFRa are in terms of cardiac maturation *in vivo*. Neither of these genes is cardiac-specific and neither has been implication in heart development *in vivo* that I know of. In addition, direct regulation of PDGFRa gene expression by Nkx2.5 has not really been disproven as is concluded by the ChIP with a limited region of the locus, since regulatory elements can be far removed from the linear location of the transcriptional start site.

VCAM1 mediates communication between the myocardium and epicardium, which is essential for heart growth. VCAM1 deficient mice die at mid-gestation due to cardiac abnormalities such as impaired ventricular septation and a severe reduction in compact layer myocardium⁴. This phenotype is also observed in α -4-integrin knockout mice⁵. Taken together these data suggest that a key role of VCAM1 is to drive myocardial growth that is dependent upon α -4-integrin signalling from the overlying epicardium. Formation of the compact myocardium is an important step in cardiomyocyte maturation and, thus, cardiomyocytes lacking VCAM1 are unlikely to mature *in vivo*. To provide this important contextual information we have added the following text to the discussion:

In the mouse, VCAM1 mediates ventricular myocardial development through interactions α -4-integrin presented on the epicardium^{57,58}.

Previous findings from ourselves and others has demonstrated that VCAM1, in the context of *in vitro* cardiac differentiation of hPSCs, is sufficient to identify cardiomyocytes and is a marker of myocardial commitment⁶⁻⁹. This is consistent with the expression pattern observed in the embryonic mouse heart^{4, 8}.

Recent work has demonstrated a key role for PDGFRA in the midline fusion of the embryonic heart tube ¹⁰. We have added this reference and changed the text in the results to include this information:

“In addition, *NKX2-5*^{-/-}-derived GFP⁺ cells retained expression of PDGFR α (Fig. 1e), a marker of cardiac progenitor cells required for heart tube formation ²⁵, normally downregulated during heart development ^{8, 26}.”

The reviewer is correct to point out that PDGFRA regulation may rely upon distal NKX2-5 binding sites. We have altered the sentence from “*Conversely, the absence of NKX2-5 binding at the PDGFRA locus suggests that NKX2-5 does not directly regulate PDGFRA expression (Supplementary Fig. 3c).*” to:

Conversely, the absence of proximal NKX2-5 binding at the PDGFRA locus suggests that **any regulatory relationship between NKX2-5 and PDGFRA is reliant upon putative NKX2-5-bound enhancers located over 250 kb from the locus (Supplementary Fig. 3c).**

Reviewer #2 (Remarks to the Author):

This manuscript by Anderson et al. examines the mechanism of how Nkx2-5 regulates cardiogenesis. The authors generate an Nkx2-5 null and heterozygous human ESC and use these lines to examine the defects that ensue due to Nkx2-5 deletion. The role of Nkx2-5 has been extensively studied during murine development but less so in the cellular differentiation of human cardiomyocyte. In that regard the ESC system provides an attractive opportunity. Although on the whole this is a well thought out and well performed study, addressing the concerns outlined below is required.

Major concerns:

1. All of the conclusions of the manuscript are drawn from two isogenic cell lines with a heterozygous or homozygous insertion of the eGFP in the Nkx2-5 locus. Although the parent cell line is well validated, the fact that no independent line of evidence is provided to validate the findings is an important weakness. This is of further concern given in vivo evidence Nkx2-5 may exhibit haploinsufficiency. It would be useful to validate the major findings of this report with another completely independent cell line. This can now be accomplished relatively easily given the high efficiency of differentiation of multiple ESC and hiPSC lines and the relative ease of generation of biallelic deletions using current genome engineering technologies.

In line with the reviewer’s suggestion, we generated an *NKX2-5* knockout in the well-characterised and widely used H9 cell line (Supplementary Fig. 1k). Flow cytometric analysis demonstrated that this cell line also had lower levels of VCAM1 and maintained PDGFR α expression (Supplementary Fig. 1l,m). Furthermore, we demonstrate that expression of *HEY2*, *IRX4*, *MYL2*, *NPPA* and *VCAM1*, is perturbed in the H9 *NKX2-5*^{-/-} cell line (Supplementary Fig. 3a). Furthermore, transcripts for *ISL1*, *FGF10* and *BMP2* are expressed more highly in H9 *NKX2-5* null cultures. Thus, the important NKX2-5 transcriptional targets, identified in the HES3 background, are dysregulated in the H9 *NKX2-5* null lines and cardiomyocyte maturation is impaired. These data suggest that the *NKX2-5* transcriptional network defined in the HES3 background is likely to be

consistent across other genetic backgrounds. We concur with the Reviewer's assessment that including this line strengthens the manuscript.

These data are described in the text:

Further, this cell surface marker phenotype is recapitulated in H9 hESCs in which *NKX2-5* has been deleted (*NKX2-5^{eGFP/del}*; Supplementary Fig. 1k,l,m).

and

Furthermore, expression of the *NKX2-5*-dependant genes *HEY2*, *IRX4*, *NPPA*, *MYL2* and *VCAM1* is reduced in H9 *NKX2-5* knockout cardiomyocytes. Heterozygosity for *NKX2-5* did not alter *IRX4*, *HEY2*, *NPPA* or *VCAM1* expression (Supplementary Fig. 3a) consistent with the similar levels of *NKX2-5* protein observed (Supplementary Fig. 1f). In addition, transcripts upregulated in H9 *NKX2-5* null cardiomyocytes included the progenitor markers *ISL1*, *FGF10* and *BMP2* (Supplementary Fig. 3a).

and

Further, the *IRX4* and *HEY2* transcription factors are also dysregulated in H9 *NKX2-5^{eGFP/del}* cardiomyocytes whilst *TBX5* was not (Supplementary Fig 3a).

The new data presented in Supplementary Figure 1k,l,m and Supplementary Figure 3a,b are shown below:

(k) Schematic of CRISPR/CAS9 mediated mutation of *NKX2-5* locus in the H9 cell line. eGFP was introduced into one *NKX2-5* allele using the targeting strategy outlined in a. The coding sequence of the second *NKX2-5* allele was deleted resulting from Non-Homologous End Joining (NHEJ) after CRISPR/CAS9 treatment.

(l) Flow cytometric analysis of H9 *NKX2-5^{-/-}* (i.e. *NKX2-5^{eGFP/del}*) cultures at day 14 of differentiation. VCAM1 expression is greatly reduced in H9 *NKX2-5* deficient cardiomyocytes, consistent with the HES3 phenotype. SIRPA expression is unperturbed in the *NKX2-5* knockout. Numbers on plots show the percentage of cells found in the quadrants indicated.

(m) Flow cytometry demonstrates that PDGFRA expression is maintained in H9 *NKX2-5^{-/-}* cultures at day 35 of culture. This phenotype is consistent with data obtained from the HES3 cell line. Numbers on plots show the percentage of cells found in the quadrant.

(a) Q-PCR analysis of H9 $NKX2-5^{eGFP/w}$ and $NKX2-5^{-/-}$ cultures at day 14 of differentiation. H9 $NKX2-5$ null cardiomyocytes show reduced transcript levels for $NKX2-5$, $HEY2$, $IRX4$, $MYL2$, $NPPA$ and increased $ISL1$, $FGF10$ and $BMP2$. Levels of $TBX5$ are consistent between H9 $NKX2-5$ heterozygotes and null cardiomyocytes. These data are match the gene expression profile observed in HES3 cells (Figure 3). Data are presented as gene expression relative to H9 $NKX2-5^{eGFP/w}$ and represent mean \pm SEM (n = 3-6).

(b) Bar graph of Q-PCR analysis $NKX2-5^{eGFP/w}$ and wildtype cardiomyocytes. Data are presented as gene expression relative to wildtype ($NKX2-5^{w/w}$) and represent mean \pm SEM (n = 3).

In addition we describe the generation of this line in the methods as follows:

CRISPR/Cas9 genome editing was used to delete the coding sequence of $NKX2-5$. Briefly, synthetic oligonucleotides containing the desired $NKX2-5$ protospacer sequences (5' guide: 5' CCATGTTCCCCAGCCCT and 3' guide: 5' GACCGATCCCACCTCAAC) and sequence overhangs compatible to the BbsI were annealed and the duplex cloned into the BbsI site of the vector pSpCas9(BB)-2A-GFP vector (PX458; Addgene Plasmid #48138)⁶³. Subsequently, H9 cells⁶⁴ in which one allele had been targeted with sequences encoding eGFP, H9 $NKX2-5^{eGFP/w}$ ¹⁸, were electroporated with the pSpCAS9(BB)-2A-GFP plasmid, and GFP-expressing single cells isolated by FACS after 2–5 days using a BD Influx cell sorter as described⁶⁵. Individual GFP-expressing clones were expanded and screened by PCR ($NKX2-5$ Fwd 5' TTGTGCTCAGCGCTACCTGCTGC and $NKX2-5$ rev 5' GGGGACAGCTAAGACACCAGG) to identify clones with modified alleles. The mutant alleles were confirmed by sequencing of the PCR products and pluripotency of the H9 $NKX2-5^{eGFP/del}$ was confirmed by expression of pluripotent stem cell markers (ECAD, SSEA-4, TRA160, CD9) and differentiation to mesodermal and endodermal lineages.

In respect to $NKX2-5$ function, we present evidence that $NKX2-5$ protein levels are comparable between $NKX2-5$ wildtype and the $NKX2-5^{eGFP/w}$ heterozygotes (Supplementary Fig 1f). We refer to this new data in the text as follows:

.... and $NKX2-5$ levels were comparable between $NKX2-5^{eGFP/w}$ and wildtype cells (Supplementary Fig. 1f).”

Supplementary Figure 1f is shown below:

f

(f) Western blot to detect NKX2-5 from wildtype ($NKX2-5^{w/w}$), $NKX2-5^{eGFP/+}$ and $NKX2-5^{-/-}$ cultures at day 14 of differentiation demonstrates that $NKX2-5$ heterozygote and wildtype $NKX2-5$ cardiomyocytes have comparable NKX2-5 levels and $NKX2-5$ null cardiomyocytes do not express NKX2-5.

In addition, we provide transcriptional evidence that key $NKX2-5$ target genes such as *HEY2*, *IRX4*, *NPPA* and *VCAM1* are expressed in $NKX2-5^{eGFP/w}$ at a level that is comparable to wildtype (Supplementary Fig. 3b, above). This result is described in the text:

Heterozygosity for $NKX2-5$ did not alter *IRX4*, *HEY2*, *NPPA* or *VCAM1* expression (Supplementary Fig. 3b) consistent with the similar levels of NKX2-5 protein observed (Supplementary Fig. 1f).

In addition, we provide further evidence against haploinsufficiency altering the phenotype by showing that GJA1 (connexin 43) levels are comparable in $NKX2-5$ wildtype and $NKX2-5$ heterozygotes (i.e. $NKX2-5^{eGFP/w}$) (Fig. 4i). These data imply that gap junction and intercalated disc formation is unaltered in $NKX2-5$ heterozygotes. These data have been included in the text as follows:

...and GJA1 protein levels were restored in HEY2::ER rescued cultures to levels comparable to both $NKX2-5^{eGFP/w}$ and wildtype cardiomyocytes (Fig. 4i).

Taken together this new data suggest heterozygosity for $NKX2-5$ does not impact on the phenotypes observed.

With regard to the Reviewer's concerns regarding $NKX2-5$ haploinsufficiency, we have previously demonstrated that the wildtype ($NKX2-5^{w/w}$) and heterozygous $NKX2-5^{GFP/w}$ derived CMs are functionally equivalent^{8, 11}. Specifically, we have demonstrated that wildtype and $NKX2-5^{eGFP/w}$ bioengineered heart tissue generate equal muscle force¹¹ and that $NKX2-5$ heterozygotes have equivalent responses to the chronotropic drugs isoprenaline and Endothelin 1 (ET1)⁸. Furthermore, electrophysiology data from the $NKX2-5^{GFP/w}$ are comparable to data that we, and others, have previously published for a number of wildtype hESC and iPSC lines¹²⁻²². These data are all consistent with the conclusion that heterozygosity for $NKX2-5$ does not significantly affect cardiomyocyte specification or function *in vitro*.

Evidence to support the notion that haploinsufficiency for $NKX2-5$ leads to impaired cardiomyocyte (CM) differentiation or function is sparse. The initial report by Schott *et al* describes three $NKX2-5$ mutations associated with heart disease all of which encode missense point mutations in the

protein, which generate dominant negative alleles, not null alleles²³. Since this initial paper, there have been 38 additional *NKX2-5* mutations reported to be associated with heart disease^{24, 25}. However, analysis of these mutants reveals that all 38 will still encode a mutant *NKX2-5* protein rather than a null allele. Not all of the 38 mutations have been fully characterized biochemically but, of those that have, it is suggested that they also act in a dominant negative manner by competing for transcriptional co-factors with the wildtype *NKX2-5* protein^{26, 27}. Therefore, we argue that true haploinsufficiency for *NKX2-5* has yet to be associated with human myocardial disease. Furthermore, the most penetrant phenotype associated with these point mutants is atrioventricular block, which affects the conduction system rather than the myocardium. The other major classes of cardiac abnormalities found in patients with *NKX2-5* mutations are likely to arise from incorrect cardiac patterning (e.g. tetralogy of Fallot, atrial septation defects), again suggesting that CM differentiation, growth and function is relatively normal in these individuals.

In mouse models, haploinsufficiency for *Nkx2-5* results in a phenotype that is mild with no conduction problems observed in males, a slightly prolonged P-R interval in females and moderately impaired atrial morphogenesis depending on genetic background²⁸. In addition, *Nkx2-5* heterozygote mice are normal and viable. No difference in expression levels of *Nkx2-5* target genes was observed in *Nkx2-5* heterozygote mice²⁹. Therefore, the animal models strongly suggest that CM growth, differentiation and function proceed normally in *Nkx2-5* heterozygous mice. In contrast, mice carrying pathogenic point mutations in *Nkx2-5* developed atrioventricular block, consistent with the clinical presentation of most patients expressing *NKX2-5* point mutations. Transgenic mice overexpressing a mutant form of *Nkx2-5* associated with congenital heart disease (*Nkx2-5*I183P) disease develop severe atrioventricular block and die of heart failure within 12 weeks³⁰. Therefore, even in the presence of two wildtype *Nkx2-5* alleles this mutant protein is capable of disrupting the conduction system in a dominant negative manner. In summary, we believe that haploinsufficiency for *NKX2-5* is highly unlikely to impact on the results presented.

2. Likewise, consideration should be given to bolstering the developmental arguments of this interaction using mouse embryonic studies.

The goal of this study was to dissect the role of *NKX2-5* in a human system to uncover species-specific functions of *NKX2-5*. In this context, further murine studies are unlikely to be informative. We point the Reviewer to our response to Points 1, 4 and 5 of Reviewer 1 for further elaboration and to highlight that we have placed our data into the developmental biology framework provided by studies in the mouse. Furthermore, we reiterate that *NKX2-5* may be deployed differently in humans than mice, as evidenced by the discovery in this paper that *HEY2* is a key downstream transcription factor of *NKX2-5*, which has not previously been reported in the mouse.

3. The authors suggest that although the *Hey2* promoter has multiple *Nkx2-5* binding sites, *Nkx2-5* may indirectly activate *Hey 2* expression (e.g. via *Coup-TF* or *BMP* signaling). Determining whether the interaction between the *Hey2* promoter and *Nkx2-5* is directly responsible for the activation of *Hey 2* promoter would be of interest and would significantly improve the manuscript. This may be done with suppression expressions of the *Coup-TF* or *BMP* signaling pathway in the PSC used in the study or alternatively using cell lines that do not express these signaling pathways and determining whether *Nkx2-5* is either necessary or sufficient for the expression of *Hey2* using a transcriptional readout of that promoter.

In line with the reviewer's suggestion, we sought to determine if NKX2-5 could activate transcription from the putative HEY2 enhancer elements we identified by ChIP-seq. In transient transfection assays in HEK 293T cells, NKX2-5, in collaboration with the established co-factors GATA4 and TBX20, was able to activate gene expression from the most proximal 5' and 3' elements (Supplementary Fig. 4f). These data show a statistically significant activation of 3 fold (5' enhancer) and 5 fold (3' enhancer) in this heterologous assay. In conjunction with the molecular data demonstrating down regulation of HEY2 in *NKX2-5* null cardiomyocytes and NKX2-5 binding to these regions our data support the hypothesis that HEY2 is directly regulated by NKX2-5.

Supplementary Figure 4f is shown below:

(f) *In vitro* transcriptional analysis using FLAG-tagged *Nkx2-5* expression vectors and a luciferase reporter containing the 5' or 3' HEY2 enhancer elements. The cardiac transcription factors *Gata4* and *Tbx20a* were also included where indicated. Data represent the mean and SEM (n=3) and values of each biological replicate are indicated by circles. **** p<0.0001.

To clarify, the NKX2-5 binding sites are located some distance away from HEY2, not in the proximal promoter, and are likely to be enhancer elements. As a result, it remains formally possible that the NKX2-5-HEY2 interaction is not direct. Hence, we provided possible alternative mechanisms, namely BMP signalling and COUP-TF regulation, in the discussion. To address the possibility that HEY2 expression levels were due to reduced BMP signalling levels we treated differentiating cultures with the BMP antagonist DMH1. These experiments show that BMP pathway is not a key regulator of HEY2 (Supplementary Fig. 4g).

Supplementary figure 4g is shown below:

(g) Bar graph of expression of *HEY2* in cardiomyocytes after BMP signaling inhibition using DMH1. *HEY2* levels are not reduced in presence of DMH1.

This new has been added to the text as follows:

Furthermore, NKX2-5, in collaboration with established transcriptional co-factors GATA4 and TBX20⁵², is able to transactivate both proximal 5' (-190 kb) and 3' (+379 kb) putative enhancer

elements in HEK 293T cells (Supplementary Fig. 4f). In some contexts, *Hey2* is induced by BMP and TGF β signaling⁵³ and expression of components of both pathways is reduced in *NKX2-5* null cardiomyocytes (Supplementary Fig. 3I), which may lead to a decrease in *HEY2*. However, *HEY2* transcript levels were not reduced when differentiating *NKX2-5*^{eGFP/w} cardiomyocytes were exposed to the BMP antagonist DHM1 (Supplementary Fig. 4g). Taken together these data support the hypothesis that *HEY2* is directly regulated by *NKX2-5*.

With respect to COUP-TF2 there is only a single report of a chemical inhibitor of this protein³¹. Thus, a well-established pharmacological approach to modifying COUP-TF2 activity is not readily available. Therefore, we have begun to examine the role of COUP-TF2 genetically, however, the results of this work lie outside the scope of the current manuscript.

Finally, we have altered the discussion to reflect our new findings as follows:

Several lines of evidence suggest the *NKX2-5*-*HEY2* regulatory relationship is direct. Firstly, *HEY2* expression in cardiomyocytes is dependent on *NKX2-5*. Secondly, while *NKX2-5* is bound at DNA elements some distance from the *HEY2* translational start site it is the only gene within 5 Mbp that is dysregulated. Thirdly, *NKX2-5*, in the presence of known co-factors GATA4 and TBX20, was able to transactivate two of these *HEY2* regulatory elements in a heterologous system. Further, inhibition of the BMP signaling pathway does not alter *HEY2* levels suggesting that in cardiomyocytes *HEY2* expression is not regulated by a BMP regulatory axis. It is likely that *HEY2* regulation is multifactorial and complex. In this context, it is interesting to note increased expression of *NR2F2* (COUP-TFII), a known repressor of *HEY2*⁴⁹, in *NKX2-5* null cardiomyocytes. Thus, *HEY2* regulation by *NKX2-5* may include an indirect component through COUP-TF-dependent repression.

Additional concerns:

1. In Figure 1, eGFP expression appears to be nuclear. Are the investigators using a nuclear GFP and not eGFP? If so, this should be made explicit. Otherwise an explanation regarding the nuclear localization of eGFP should be provided.

To clarify, eGFP was used. The nuclear enrichment is an artefact of fixation and a low level of eGFP remains visible throughout the cardiomyocytes. The supplementary movies show widespread eGFP expression throughout the cardiomyocytes. In addition, multiple reports with the *NKX2-5*^{eGFP/w} line have shown eGFP is not restricted to the nucleus in live cells, e.g.^{6, 8, 11, 20, 32-34}.

2. Figure 1D: Is the qPCR performed in FACS sorted eGFP+ cells? This data seems to suggest that eGFP+ cells are very heterogeneous. It would seem that this heterogeneity may undermine some of the conclusions that the authors are drawing and this should be investigated—are the eGFP low cells at different developmental/differentiation stage than the high population, is *Nkx2-5* expression (at least in the het cell line) correspondingly lower? Which population is driving the observations from this study?

To clarify, Figure 1D was performed in order to establish that the contractile *NKX2-5* null cultures expressed cardiomyocyte markers and did not use FACS sorted cells. The heterogeneity in GFP+ expression is due to the expression of *NKX2-5* in a range of cardiovascular cell types including smooth muscle, endothelial cells⁶, atrial cells³⁵ and sinoatrial node precursors²⁰. To alleviate this concern we focused on the GFP high population (i.e. cardiomyogenic) in both genotypes for molecular profiling. Thus, our data are based on the cell population that expresses high levels of

GFP from the NKX2-5 locus, which circumvents potential difficulties in ascribing phenotype to genotype raised by the reviewer.

3. Figure 1G: It appears that there are two different eGFP+ populations, an eGFP high and low population. Can the authors provide any explanations of these findings and/or an analysis of their significance?

These FACS plots show a relatively early differentiation time point (day 14). The low GFP population will comprise progenitors and other cell types that are GFP low such as endothelial cells, possibly fibroblasts and smooth muscle, conduction cells specified from the NKX2-5 positive population^{6, 20, 21, 35, 36}. To address this concern we have added the following text to the manuscript: ...both NKX2-5^{eGFP/w} and NKX2-5 null GFP positive populations were heterogeneous, with low GFP expressing cells representing cardiac precursors and non-myocytes (Supplementary Fig. g)²⁰⁻²⁴.

4. Figure 2A: The Nkx2-5^{-/-} Fluo4-AM tracing appears to be very noisy with little discernable signal. While this may be due to inherent defects in calcium handling in these cell types, the poor signal to noise ratio in these cells makes it difficult to assess the coordination of calcium flux in adjacent cells. Perhaps the ^{-/-} cells are simply more sensitive to the known cytotoxic effects of calcium dyes.

We have examined the Fluo4-AM tracing data from all experiments and there is consistent signal in the NKX2-5 null cardiomyocytes suggesting that NKX2-5^{-/-} cells are not more sensitive to calcium dyes. In part, the noisy appearance of this data is due to the reduced amplitude of calcium flux in NKX2-5^{-/-} cardiomyocytes (Fig. 2c). Calcium imaging was used to quantify the contractile coordination observed in NKX2-5 null cardiac monolayers as shown in Supplementary Movies 1 and 3. In this context, the data support the initial observation that contractility was perturbed in NKX2-5 cardiac cultures. Furthermore, induction of NKX2-5 and HEY2 using the GAPTrap system was able to restore synchronous contractility, and VCAM1 and GJA1 expression (Fig. 4g,h,i). It is important to note that all synchronicity assays using Fluo4 were performed blind to genotype, which gives a greater degree of confidence in the findings.

5. The contractility analysis of the organoids is superficial. The investigators need to define whether the apparent defect in contractility is due to abnormal electromechanical coupling (as the data would seem to suggest) or to an actual defect in cardiomyocyte contractile function--It is possible that the functional difference between the two cell types is only due to poor electrical coupling of the organoids and not to a contractility defect. This needs to be sorted out potentially through single cell analysis of contractility (multiple options from multiple groups available for this type of analysis).

As outlined in the molecular analysis, both myogenesis and components of the electrical system are impaired in NKX2-5 knockouts (Fig 3). The organoid data is provided to complement and support the extensive molecular and cellular data characterising the defective cardiomyogenic program of NKX2-5 null cardiomyocytes. In this context, this data is not to be viewed in isolation and demonstrates that muscle force generation is compromised in NKX2-5 null cultures. As the reviewer correctly points out this compromised performance in the organoid system may arise from deficiencies in electrical coupling within the bioengineered tissue. In the future, it will be important to dissect the mechanism of reduced force using single cell methodologies as suggested. However, this would extend beyond the scope of the existing work.

6. Figure S1. D-too many cells are off the scale

As the reviewer suggested we have repeated this experiment to maintain the signal within the detection limit of the flow cytometer. This new data is now shown in Supplementary Figure 1d and altered the figure legend accordingly:

d

(d) Flow cytometry demonstrating that $NKX2-5^{-/-}$ hESCs retain expression of pluripotency markers TRA-1-60, CD9, EPCAM and SSEA4. Numbers on each plot show the percentage of cells found in that quadrant. These data demonstrate that 98 per cent of $NKX2-5$ null hESCs express these markers.

7. Figure S1E- cellular constructs not clear, example seems to be of an empty well.

This figure contained sections of teratomas derived from $NKX2-5^{-/-}$ with examples of derivatives of all three germ layers. The reviewer may be referring to Supplementary Figure 2e, which contained a bright field image of the cardiac organoids used to study contractile force. In order to clarify this, we amended the figure to include an epifluorescence image showing staining for ACTN2 and nuclei counterstained with DAPI. Furthermore, the polymer uprights around which the organoids coalesce are now indicated in the figure. Movement of these uprights is used to calculate contractile force. The new figure and legend are shown below:

e

(e) Bright field and epifluorescence images of bioengineered cardiac organoids used to assess contractile force. Epifluorescent image captures cardiac α -actinin (ACTN2) and nuclear DAPI staining. U = polymer uprights around which cardiac organoids form. Scale bar = 500 μ M”

Reviewer #3 (Remarks to the Author):

Review for NKX2-5 regulates human cardiomyogenesis via a HEY2-dependent transcriptional network” (Anderson et al).

To determine a molecular mechanism by which the cardiac transcription factor NKX2-5 functions in developing cardiomyocytes, Anderson et al depleted the wild type allele of NKX2-5 in a hESC model in which one copy of NKX2-5 was replaced with GFP. NKX2-5^{-/-} cells differentiate into cardiomyocytes but have delayed onset of differentiation. NKX2-5^{-/-} iCMs also retain PDGFR α expression and have reduced VCAM1 expression, suggesting perturbed differentiation.

The authors then performed contractility analyses to demonstrate impaired electrophysiology in NKX2.5^{-/-} cells. These cells demonstrate abnormal contraction as shown by calcium oscillation calculations, MEA analysis, and patch clamp analysis. They then generated bioengineered cardiac organoids from these cells to determine that mutations in NKX2-5 exhibited decreased contractile force.

To determine the NKX2.5 gene network the authors performed RNA-seq analysis with ChIP-seq analysis to determine direct targets of NKX2.5. RNA seq analysis determined that ventricular specific genes were reduced in the NKX2.5 knockout cells while cardiac progenitor genes were upregulated in the knockout cells. ChIP-seq analysis showed that NKX2.5 was bound near genes involved in cardiac development. When ChIP peak data was overlaid with RNA-seq expression data it was found that genes that are directly bound and activated by NKX2.5 are involved in channel activities, and genes that are directly bound and repressed by NKX2.5 are involved in calcium binding and neurogenesis. The authors then examined gap junction proteins to explain differences in ion channel expression and found that gap junction GJA1 was not localized to the gap junctions. They also noted aberrant expression of genes required for the SHF, AVC, and OFT. The authors then determined the transcription factor network that interacts with NKX2.5 and found a number of transcription factors (including HEY-2 and the IRX cluster) that are misregulated in NKX2.5 null iCMs. They then performed a rescue assay using GAPTrap and found that temporal introduction of NKX2.5 and HEY2, but not IRX4, rescued the null phenotype in terms of VCAM1 expression, suggesting that HEY2 is one of the critical mediators of the NKX2.5-dependent transcriptional network.

Major issues

1) Figure 1: a) The hESCs seem to differentiate fairly well in culture, with >80% of the cells expressing GFP and ACTN2 by day 10 of differentiation. In supplementary Figure 1H, cell sorting analysis was performed to characterize ACTN2 and GFP expression in culture at Day 14. Although many of the cells (please provide quantitation here) are ACTN2⁺/GFP⁺, and some cells that are ACTN2⁻/GFP⁻, there are also a significant number of cells that are ACTN2⁺,GFP⁻. Please comment on why that might be and how these cells may affect your analysis.

We provide quantitation of the ACTN2⁺/GFP⁺ population in Figure 1c (right hand graph). With respect to the presence and composition of the ACTN2⁺/GFP⁻ population, there is a biological and technical basis for identification of this population by flow cytometry. Firstly, the biological basis is that a subset of the cells will be cardiac progenitors that are now committing to other fates such as smooth muscle⁶ or cells of the conduction system^{20, 21}. Two confounding technical factors affect detailed analysis of this population as some of these cells are likely to have lost GFP during the fixation and permeabilization process and with intracellular FACS for myofibrillar proteins it is our experience there is some cross reactivity with smooth muscle cells. Therefore, it is possible that ACTN2⁺/GFP⁻ cells may result from cross reactivity of ACTN2 antibody with other myogenic cell types. As the ACTN⁺/GFP⁻ population is similar in both NKX2-5^{eGFP/w} and NKX2-5^{-/-} and may include a large number of cells due to technical reasons (i.e. loss of GFP signal and cross reactivity) we do not believe the ACTN2⁺/GFP⁻ population alters our findings or analysis.

2) All comparisons in the manuscript are between the heterozygous and homozygous null cell. Congenital heart disease is caused by haploinsufficiency of NKX2-5. Why are there no comparisons of heterozygous mutant cells to wild-type HES3 cells? For example, how many NKX2-5 binding sites differ between wildtype and heterozygous cells? How many differences in RNA expression?

We present additional data showing that NKX2-5 protein levels are comparable between *NKX2-5* wildtype and the *NKX2-5^{eGFP/w}* heterozygotes (Supplementary Fig 1f) and the gene expression of key *NKX2-5* markers is not perturbed in *NKX2-5^{eGFP/w}* (Supplementary Fig. 3b). Furthermore, we demonstrate that GJA1 (Connexin 43) levels are comparable in *NKX2-5* wildtype and *NKX2-5* heterozygotes (i.e. *NKX2-5^{eGFP/w}*) (Fig. 4i). These data are described in the text as follows:

.... and NKX2-5 levels were comparable between *NKX2-5^{eGFP/w}* and wildtype cells (Supplementary Fig. 1f).”

f

(f) Western blot to detect NKX2-5 from wildtype (*NKX2-5^{w/w}*), *NKX2-5^{eGFP/+}* and *NKX2-5^{-/-}* cultures at day 14 of differentiation demonstrates that *NKX2-5* heterozygote and wildtype *NKX2-5* null cardiomyocytes have comparable NKX2-5 levels and *NKX2-5* null cardiomyocytes do not express NKX2-5.

and

...and GJA1 protein levels were restored in HEY2::ER rescued cultures to levels comparable to both *NKX2-5^{eGFP/w}* and wildtype cardiomyocytes (Fig. 4i).

i

(i) Western blot showing restoration of GJA1 (connexin 43) levels by HEY2 and that wildtype (HES3) and *NKX2-5^{eGFP/w}* GJA1 levels are comparable.

Furthermore, as outlined in response to Reviewer’s 1 (point 1) and 2 (point 2), heterozygosity for NKX2-5 does not alter electro-mechanical function or chronotropic drug response of PSC derived cardiomyocytes^{8, 11, 16 19-22} and evidence for haploinsufficiency resulting from NKX2-5 loss of function alleles has not been conclusively proven in patients with congenital heart disease. The issue of *NKX2-5* haploinsufficiency is addressed in the detailed responses to Reviewers 1 and 2.

These data and the evidence outlined above in response to Reviewer 2 (Point 1) strongly suggest that haploinsufficiency for NKX2-5 is unlikely to impact on our results.

We purposefully chose to use *NKX2-5^{eGFP/w}* in order to allow flow cytometric enrichment of hESC derived cardiomyocytes, and, therefore, it is difficult to directly compare wildtype cells as they lack a GFP marker in the *NKX2-5* locus used for enrichment. Given that the evidence for heterozygosity for *NKX2-5* impacting on cardiomyocyte differentiation is scarce (Reviewer 2 (Point 1)) and our new data providing additional evidence for the functional equivalence between *NKX2-5* wildtype and heterozygotes, we believe our comparison of *NKX2-5* null with *NKX2-5^{eGFP/w}* cardiomyocytes is a valid approach.

3) Fig 3. Differences in RNA expression between heterozygous and homozygous NKX2-5 cells and comparison to NKX2-5 CHIP-Seq peaks. Again, this data should also be compared to wildtype cells. Do heterozygous cells have the same NKX2-5 binding peaks as wildtype cells?

While it would not be surprising to see some minor differences in NKX2-5 binding between heterozygous and wildtype cells, in the context of an *in vitro* differentiation system, and as described above in response to Reviewers 1 and 2, such differences, if they exist, do not lead to any observable phenotypic differences. Further, our new data (Supplementary fig. 1f and Fig. 4i) show that NKX2-5 and connexin 43 protein levels are comparable between the two genotypes. Given these findings, and previous data (both ours and others) showing phenotypic equivalence between wildtype and *NKX2-5^{eGFP/w}* cardiomyocytes^{8, 11, 16, 19-22}, we believe that the genomic occupancy pattern of NKX2-5 is unlikely to vary significantly between *NKX2-5* wildtype and heterozygotes.

4) RNA-seq analysis for GFP+ enriched cardiomyocytes at Day 10 of differentiation: PDGFR α is shown to be enriched in NKX2.5 null cells, but earlier in the text PDGFR α is reported as unchanged up to day 14. Please clarify.

This discrepancy is due to the fact that this data measure two different read-outs. The transcript is different but PDGFR α protein levels at the cell surface are comparable at this stage of *in vitro* differentiation. This data is consistent with a number of studies showing a low level of concordance between transcript and protein levels³⁷⁻³⁹. Nevertheless, the increased *PDGFRA* transcript levels are manifest as increased PDGFR α protein levels at day 42 of differentiation (Fig. 1E, F, Supplementary Fig. 1m).

5) Materials and methods paragraph for the GAPTrap assay seems to be missing.

In line with the reviewer's suggestion we have added this information to the materials and methods as follows:

GAPTrap Rescue Experiments.

We utilised the GAPTrap strategy to engineer cell lines expressing transcription factors of interest from the GAPDH locus, as described previously⁵⁴. GAPTrap targeting vectors were modified such that sequences encoding NKX2-5, HEY2 and IRX4, all fused in-frame to the ligand-binding domain of the human Estrogen Receptor⁷⁵, were placed immediately 3' of the T2A peptide cleavage signal (Supplementary Figure 4). All cloning was performed using the InFusion HD cloning system

(TaKara, 638910). To induce nuclear translocation of the NKX2-5::ER, HEY2::ER and IRX4::ER fusion proteins the ligand 4-Hydroxytamoxifen (Sigma, T176) was added to cell culture media at final concentration of 0.5 mM. For rescue experiments, 4-Hydroxytamoxifen (4-OHT) was added from differentiation day 5 onwards.

6) NKX2.5, HEY2, and IRX4 are induced by adding 4-OHT at day 4. Why does the VCAM-APC expression increase so drastically before induction of these proteins for the rescue experiment in supplemental figure 4h?

These data were collected on day 14, which we now make clear on Supplementary Fig. 4h and in the figure legend. We apologise for the unclear presentation of this data. To address this concern, we have modified Supplementary Figure 4j such that the Y-Axis is now labelled “Day 14 VCAM-APC Geo. Mean”. Further, in line with the reviewer’s suggestion, we have added this information to the figure legend as follows:

(j) Bar graphs showing quantification of flow cytometric analysis of day 14 VCAM1 expression (see Fig. 4F). The day of 4 OHT addition is indicated on the X-axis. Geo. Mean = geometric mean fluorescence. Data represent mean \pm SEM (n = 4). *** p<0.001.

7) In the discussion NKX2.5-HEY2 regulatory relationship is postulated to be indirect. This should be tested by coimmunoprecipitation experiments in the iCMs. Additionally, transcriptional assays should be performed to test this hypothesis. Are HEY2 binding motifs upregulated near NKX2.5 binding sites?

To clarify, our data suggests NKX2-5 directly regulates *HEY2* via the two-enhancer elements identified. It is clear that NKX2-5 binds at sites that flank the *HEY2* locus and that no other genes within 5 Megabases of these sites are dysregulated. In line with the reviewers suggestion we provide new data demonstrating the NKX2-5, in collaboration with known co-factors TBX20 and GATA4, can transactivate transcription from these enhancers in a heterologous assay. This data is presented in Supplementary figure 4f and is shown below:

(f) *In vitro* transcriptional analysis using FLAG-tagged Nkx2-5 expression vectors and a luciferase reporter containing the 5' or 3' HEY2 enhancer elements. The cardiac transcription factors Gata4 and Tbx20a were also included where indicated. Data represent the mean and SEM (n=3) and values of each biological replicate are indicated by circles. **** p<0.0001.

We do not hypothesize or investigate whether NKX2-5 and HEY2 are co-factors and, as such, the establishing a biochemical interaction by co-immunoprecipitation between these two proteins lies outside the scope of this manuscript.

With respect to enrichment of HEY2 binding motifs at NKX2-5 binding sites we did not identify HEY2 binding motifs as being enriched within the CHIP-seq data set of NKX2-5 bound sites.

Minor issues

Supplementary Figure 1C: The karyotype image is too small to actually determine if any defects in karyotype are present.

Karyotyping was performed by the Cytogenetics service at the Monash Medical Centre. A total of 20 metaphase spreads per cell line were analysed and no abnormalities were recorded. As the reviewer points out the illustrative karyotyping image is small in order to fit into the space constraints, however, gross rearrangements would be visible. Furthermore, we have characterised the karyotypes of all genotypes by high density SNP profiling and added this to the methods as follows:

“Genomic integrity of selected genetically modified lines was assessed either using the Illumina HumanCytoSNP-12 v2.1 array at the Victorian Clinical Genetics Service, Royal Children’s Hospital (Melbourne)”

Thus, we provide multiple lines of evidence that the modified cell lines remain chromosomally normal.

Supplementary Figure 1: For all flow cytometry analyses, please provide quantitation for your data.

Supplementary Figure 1 is provided to demonstrate that the cells have maintained pluripotency markers after genetic modification. This data in conjunction with the capacity to form all three germ layers (Supplementary Figure 1d) and the capacity to generate cardiac mesoderm (Supplementary Figures 1f,g) demonstrate that pluripotency is maintained. Quantification of this data is now provided on the plots with the percentage of CD9/EPCAM double positive, SSEA4 positive and TRA-1-60 positive cells indicated. We have altered the figure legend to read

(d) Flow cytometry demonstrating that $NKX2-5^{-/-}$ hESCs retain expression of pluripotency markers TRA-1-60, CD9, EPCAM and SSEA4. Numbers on each plot show the percentage of cells found in that quadrant. These data demonstrate that 98 per cent of NKX2-5 null hESCs express these markers.

The new data are shown below

d

Supplementary Figure 1g and h are representative plots of the flow cytometric profiles used to generate the quantification data in Figure 1c. These findings were described in the manuscript as follows “When differentiated to the cardiac lineage as monolayers, $NKX2-5^{-/-}$ hESCs formed GFP⁺ cells with similar kinetics to the parental $NKX2-5^{eGFP/w}$ line and, by day 14 of differentiation, both cultures contained similar proportions of GFP⁺ and ACTN2⁺ cells (Fig. 1b, c and Supplementary Fig. 1g, h)”. To clarify this relationship we have amended the text to read “... (Fig. 1b, c and **see Supplementary Fig. 1g, h for representative flow cytometry plots**)”

We have added quantification data to the new plots shown to demonstrate that the VCAM1 and PDGFRA phenotypes are consistent in H9 $NKX2-5$ null. This data is shown below:

(k) Schematic of CRISPR/Cas9 mediated mutation of $NKX2-5$ locus in the H9 cell line. eGFP was introduced into one $NKX2-5$ allele using the targeting strategy outlined in **a**. The coding sequence of the second $NKX2-5$ allele was deleted resulting from Non-Homologous End Joining (NHEJ) after CRISPR/Cas9 treatment.

(l) Flow cytometric analysis of H9 $NKX2-5^{-/-}$ (i.e. $NKX2-5^{eGFP/del}$) cultures at day 14 of differentiation. VCAM1 expression is greatly reduced in H9 $NKX2-5$ deficient cardiomyocytes, consistent with the HES3 phenotype. SIRPA expression is unperturbed in the $NKX2-5$ knockout. Numbers on plots show the percentage of cells found in the quadrants indicated.

(m) Flow cytometry demonstrates that PDGFRA expression is maintained in H9 $NKX2-5^{-/-}$ cultures at day 35 of culture. This phenotype is consistent with data obtained from the HES3 cell line. Numbers on plots show the percentage of cells found in the quadrant.

Figure 2A-B: Please specify the time point for the cells used in this analysis.

We have added this information (red text) to the figure legend.

“(a) Representative graphs showing co-ordination of calcium flux in **day 16** cardiomyocyte monolayers derived from $NKX2-5^{eGFP/w}$ and $NKX2-5^{-/-}$ hESCs as detected by Fluo4-AM.”

In addition, we have added the differentiation day (red text) of cell seeding to the Materials and Methods such that it now reads:

“Differentiated cells (**Day 10**) were seeded onto Geltrex coated optical tissue culture treated 96 well plates at 1.5×10^4 cells/cm². Cells were analysed 4-6 days post plating.”

The addition of this information clarifies the time point used for analysing Ca²⁺ kinetics.

Why did you perform the ChIP analyses using cells at Day 10 when all other analyses were performed at Day 14?

This time point for ChIP-seq was chosen to be consistent with the RNA-seq data. Day 10 was chosen as it was permissive for the FACS isolation of sufficient quantities high GFP+ve cardiomyocytes (see response to Point 1, also Supplementary Fig. 1g). Day 14 cultures are established contractile cultures, which facilitated the functional studies reported in Figure 2 and Figure 4.

Figure 3i. It is customary to place a black line on a western blot designating a break in the image. Please also include the molecular weights or provide an unaltered image.

In line with the reviewer's suggestion we have added the black line to denote a break in the plot in Figure 3i and denoted the position of the molecular weight markers. This figure is shown below:

I

(i) Western blot detection of GJA1 in *NKX2-5^{eGFP/w}* and *NKX2-5^{-/-}* cultures confirms reduction in GJA1 observed in (h). Size markers in kDa are indicated to the left of the blot.

There are a number of typos in the results and the materials and methods section. Please correct typos accordingly.

We have thoroughly reviewed the methods section in line with the reviewer's suggestion.

Page 16. Please provide catalog information for PE/Cy7 conjugated secondary antibody.

We have provided the catalogue number (405315).

There are many instances in which the figures that are referenced in the results section do not correspond with the text. For example, on page 5, for your MEA analysis, the basal rate of contraction is shown in Figures 2d and 2e, and the prolonged field potential is shown in Figures 2d and 2f. Please fix the references to your figures in the second paragraph under the "Perturbed electrophysiology..." header.

We have corrected the manuscript in line with the reviewer's suggestions.

Please correct the references to figures throughout the manuscript.

In particular Figure 3F-G, ion channel and transporter genes had altered expression in *NKX2-5* cells, and that differential expression was maintained throughout differentiation. However, none of

the genes highlighted in figure 3F are reported in 3G, and vice versa. In addition to *KCNH2a/2b*, please show the RNA-seq expression analysis in 3G with the genes highlighted in 3F (*SLC47A1*, *GRIA1*, etc).

We apologise for the presentation and in-text referencing of this data. To clarify the representation of this data, we have altered the sentence

“Further investigation of ion channel and transporter genes identified altered expression in NKX2-5 null cells, and Q-PCR demonstrated that differential expression was maintained throughout differentiation, as was the case for VCAM1 and PDGFRA (Fig. 3f,g)”

to read

Further investigation of ion channel and transporter genes identified a subset with altered expression profiles in *NKX2-5* null cells (Fig. 3f, Supplementary Table 1). Q-PCR during a time course of differentiation (day 7 to 42) on a subset of genes including ion channels (*SCN5A*, *KCNH2b*), cell surface markers (*VCAM1*, *PDGFRA*) and myofilament genes (*MYL2*, *MYH11*) demonstrated that differential expression for these genes was maintained throughout differentiation (Fig. 3g).

We have corrected the in-text referrals to figure 3 in the paragraph beginning “*As well as electrophysiological abnormalities...*”. Furthermore, we have double-checked the manuscript to ensure figure referrals are consistent with the data being discussed throughout the manuscript. We thank the reviewer for bringing this to our attention.

The same is true for transcription factor expression (Figure 4b). *GATA4* and *TBX5* expression is not dependent on *NKX2.5* and reference figure 4B, but this statement is not supported by the data presented in Figure 4b.

To clarify in the submitted manuscript we state:

“Expression of most cardiac GRN members, including GATA4 and TBX5, was not dependent on NKX2-5 (Fig. 3b)”.

Thus, this data is correctly referenced and supported by the data presented.

References: Response to reviewers

1. Luna-Zurita, L. *et al.* Complex Interdependence Regulates Heterotypic Transcription Factor Distribution and Coordinates Cardiogenesis. *Cell* **164**, 999-1014 (2016).
2. Bouveret, R. *et al.* NKX2-5 mutations causative for congenital heart disease retain functionality and are directed to hundreds of targets. *Elife* **4** (2015).
3. Solan, J.L. & Lampe, P.D. Connexin43 phosphorylation: structural changes and biological effects. *Biochem. J.* **419**, 261-272 (2009).
4. Kwee, L. *et al.* Defective development of the embryonic and extraembryonic circulatory systems in vascular cell adhesion molecule (VCAM-1) deficient mice. *Development* **121**, 489-503 (1995).
5. Yang, J.T., Rayburn, H. & Hynes, R.O. in *Development*, Vol. 121 549-560 (1995).
6. Skelton, R.J. *et al.* SIRPA, VCAM1 and CD34 identify discrete lineages during early human cardiovascular development. *Stem Cell Res* **13**, 172-179 (2014).
7. Uosaki, H. *et al.* Efficient and scalable purification of cardiomyocytes from human embryonic and induced pluripotent stem cells by VCAM1 surface expression. *PLoS ONE* **6**, e23657 (2011).
8. Elliott, D.A. *et al.* NKX2-5(eGFP/w) hESCs for isolation of human cardiac progenitors and cardiomyocytes. *Nat Methods* **8**, 1037-1040 (2011).
9. Wang, G. *et al.* Modeling the mitochondrial cardiomyopathy of Barth syndrome with induced pluripotent stem cell and heart-on-chip technologies. *Nat Med* **20**, 616-623 (2014).
10. Bloomekatz, J. *et al.* Platelet-derived growth factor (PDGF) signaling directs cardiomyocyte movement toward the midline during heart tube assembly. *Elife* **6** (2017).
11. Voges, H.K. *et al.* Development of a human cardiac organoid injury model reveals innate regenerative potential. *Development* **144**, 1118-1127 (2017).
12. Graichen, R. *et al.* Enhanced cardiomyogenesis of human embryonic stem cells by a small molecular inhibitor of p38 MAPK. *Differentiation* **76**, 357-370 (2008).
13. He, J.-Q., Ma, Y., Lee, Y., Thomson, J.A. & Kamp, T.J. in *Circ Res*, Vol. 93 32-39 (2003).
14. Kehat, I. *et al.* in *J Clin Invest*, Vol. 108 407-414 (2001).
15. Mummery, C. *et al.* Differentiation of human embryonic stem cells to cardiomyocytes: role of coculture with visceral endoderm-like cells. *Circulation* **107**, 2733-2740 (2003).
16. Phelan, D.G. *et al.* ALPK3-deficient cardiomyocytes generated from patient-derived induced pluripotent stem cells and mutant human embryonic stem cells display abnormal calcium handling and establish that ALPK3 deficiency underlies familial cardiomyopathy. *Eur Heart J* **37**, 2586-2590 (2016).
17. Satin, J. *et al.* Calcium Handling in Human Embryonic Stem Cell-Derived Cardiomyocytes. *Stem Cells* **26**, 1961-1972 (2008).
18. Satin, J. *et al.* Mechanism of spontaneous excitability in human embryonic stem cell derived cardiomyocytes. *J Physiol (Lond)* **559**, 479-496 (2004).
19. Dubois, N.C. *et al.* SIRPA is a specific cell-surface marker for isolating cardiomyocytes derived from human pluripotent stem cells. *Nat Biotechnol* **29**, 1011-1018 (2011).
20. Birket, M.J. *et al.* Expansion and patterning of cardiovascular progenitors derived from human pluripotent stem cells. *Nat Biotechnol*, 1-12 (2015).
21. Protze, S.I. *et al.* Sinoatrial node cardiomyocytes derived from human pluripotent cells function as a biological pacemaker. *Nat Biotechnol* **35**, 56-68 (2017).
22. Laksman, Z. *et al.* Modeling Atrial Fibrillation using Human Embryonic Stem Cell-Derived Atrial Tissue. *Sci Rep* **7**, 5268 (2017).
23. Schott, J.J. *et al.* Congenital heart disease caused by mutations in the transcription factor NKX2-5. *Science* **281**, 108-111 (1998).
24. Elliott, D., Kirk, E., Schaff, D. & Harvey, R. NK-2 Class Homeodomain Proteins: Conserved Regulators of Cardiogenesis, in *Heart Development and Regeneration*, Vol. 2, Edn. 1 569-598 (Academic Press, 2010).

25. Elliott, D.A., Kirk, E.P., Schafft, D. & Harvey, R.P. NK-2 class homeodomain proteins: conserved regulators of cardiogenesis, in *Heart Development and Regeneration*, Vol. 2. (eds. N. Rosenthal & R.P. Harvey) 571-599 (Elsevier, 2010).
26. Kasahara, H. *et al.* in *J Clin Invest*, Vol. 106 299-308 (2000).
27. Kasahara, H. & Benson, D.W. in *Cardiovasc Res*, Vol. 64 40-51 (2004).
28. Biben, C. *et al.* Cardiac septal and valvular dysmorphogenesis in mice heterozygous for mutations in the homeobox gene *Nkx2-5*. *Circ Res* **87**, 888-895 (2000).
29. Jay, P.Y. *et al.* in *FASEB J*, Vol. 19 1495-1497 (2005).
30. Wakimoto, H. *et al.* in *Genesis*, Vol. 37 144-150 (2003).
31. Le Guevel, R. *et al.* Inactivation of the Nuclear Orphan Receptor COUP-TFII by Small Chemicals. *ACS Chem Biol* **12**, 654-663 (2017).
32. Skelton, R.J. *et al.* CD13 and ROR2 Permit Isolation of Highly Enriched Cardiac Mesoderm from Differentiating Human Embryonic Stem Cells. *Stem Cell Reports* **6**, 95-108 (2016).
33. Kempf, H. *et al.* Controlling expansion and cardiomyogenic differentiation of human pluripotent stem cells in scalable suspension culture. *Stem Cell Reports* **3**, 1132-1146 (2014).
34. Bellin, M. *et al.* Isogenic human pluripotent stem cell pairs reveal the role of a *KCNH2* mutation in long-QT syndrome. *EMBO J* **32**, 3161-3175 (2013).
35. Devalla, H.D. *et al.* Atrial-like cardiomyocytes from human pluripotent stem cells are a robust preclinical model for assessing atrial-selective pharmacology. *EMBO Mol Med* **7**, 394-410 (2015).
36. Raynaud, C.M. *et al.* Human embryonic stem cell derived mesenchymal progenitors express cardiac markers but do not form contractile cardiomyocytes. *PLoS ONE* **8**, e54524 (2013).
37. Fu, X. *et al.* Estimating accuracy of RNA-Seq and microarrays with proteomics. *BMC Genomics* **10**, 161 (2009).
38. Vogel, C. & Marcotte, E.M. Insights into the regulation of protein abundance from proteomic and transcriptomic analyses. *Nat Rev Genet* **13**, 227-232 (2012).
39. Lundberg, E. *et al.* Defining the transcriptome and proteome in three functionally different human cell lines. *Mol Syst Biol* **6**, 450 (2010).

Reviewers' Comments:

Reviewer #1:

Remarks to the Author:

The revised manuscript is responsive to the original critiques and is improved. The electron micrographs are a nice addition.

Minor comment

Additional information should be provided on the methods for cardiac organoids in terms of source material and general culture conditions. At least some minimal information is needed in order for a reader to know how the experiments were done and to interpret resulting data.

Reviewer #2:

Remarks to the Author:

In this revised manuscript, Anderson et al. examine the mechanism of how Nkx2-5 regulates cardiogenesis. Overall the current iteration of the manuscript represents a marked improvement over the original submission and substantively advances the field. A few additional concerns remain and should be addressed.

1. The generation of independent cell lines using the well validated H9 line is an improvement. This portion of the studies should be expanded to include functional characterization of the H9 derived cardiomyocytes as in Figure 2.
2. The investigators argue that examination of the role of Hey2 in mouse embryos would be outside the scope of the current manuscript. While this point may be reasonable, in the absence of studies in embryos, the investigators should use the term "cellular differentiation" in lieu of "development" as the former more accurately reflects the scope of the current study.
3. The investigators use a heterologous system to show that Nkx (along with Tbx5 and GATA4) can activate Hey 2 transcription. The data shown shows relative changes in Hey expression compared to untransfected cells. It is critical to show on the same graph with the same scale the relative Hey2 expression levels in cardiomyocytes to ensure that the effect that is observed is in the physiologically relevant range.
4. The reviewers should explicitly acknowledge in the discussion that the observed contractility defects may not be primary and may in fact be all secondary to conduction defect. Without the single cell analysis, it is simply not possible to conclude that there is a primary contractility defect.

Reviewer #3:

Remarks to the Author:

The authors have addressed my concerns by indicating that the current cell culture model provides little (or no insight) into the mechanism by which haploinsufficiency of NKX2-5 causes disease. The authors have adequately responded to other, noted, concerns.

NCOMMS-17-01534: Response to Reviewers.

We thank the reviewers for their second review of the manuscript. We have addressed the reviewers concerns (blue text) as outlined below:

Reviewer #1 (Remarks to the Author):

The revised manuscript is responsive to the original critiques and is improved. The electron micrographs are a nice addition.

Minor comment

Additional information should be provided on the methods for cardiac organoids in terms of source material and general culture conditions. At least some minimal information is needed in order for a reader to know how the experiments were done and to interpret resulting data.

We have added the detailed methodology for cardiac organoid formation to the methods section.

Reviewer #2 (Remarks to the Author):

In this revised manuscript, Anderson et al. examine the mechanism of how Nkx2-5 regulates cardiogenesis. Overall the current iteration of the manuscript represents a marked improvement over the original submission and substantively advances the field. A few additional concerns remain and should be addressed.

1. The generation of independent cell lines using the well validated H9 line is an improvement. This portion of the studies should be expanded to include functional characterization of the H9 derived cardiomyocytes as in Figure 2.

We note that we have provided evidence the H9 *NKX2-5* null cardiomyocytes have reduced *HEY2*, *IRX4*, *MYL2* and *NPPA* expression (Supplementary Fig. 3a) and impaired cardiac maturation as assessed by *VCAM1* and *PDGFR α* expression (Supplementary Fig. 1l,m and Supplementary Fig. 3a). In the context of our major finding that *HEY2* is a genetic target of *NKX2-5* this data is sufficient to support the key contentions of this manuscript.

2. The investigators argue that examination of the role of *Hey2* in mouse embryos would be outside the scope of the current manuscript. While this point may be reasonable, to in the absence of studies in embryos, the investigators should use the term "cellular differentiation" in lieu of "development" as the former more accurately reflects the scope of the current study.

At no stage did we claim, or intend to claim, that the system we use equates to human embryonic development. Rather it is a model that can be used to understand the genetic networks that drive human cardiomyocyte differentiation and by extension have potential implication for human heart development. This is consistent with multiple studies in the mouse in which *in vitro* studies using ESCs recapitulate aspects of mouse development.

Therefore, we are fully aware of the caveats applying to differentiating pluripotent stem cell systems and have been careful to ensure our data is presented in such a way as to not be misleading. Nevertheless, to address this reviewer's concerns we have made changes where the term "development" may have been construed to imply a finding derived from embryology. These are as follows:

To understand how human NKX2-5 regulates myocardial development...

has been changed to

*To understand how human NKX2-5 regulates myocardial **differentiation**...*

NKX2-5-dependent transcriptional network that guides cardiomyocyte development

has been changed to

*NKX2-5-dependent transcriptional network that guides cardiomyocyte **differentiation***

NKX2-5-dependant factor for human ventricular muscle development ...

has been changed to

*NKX2-5-dependant factor for human ventricular muscle **differentiation** ...*

Fig. 1. NKX2-5 regulates cardiomyocyte development

has been changed to

*Fig. 1. NKX2-5 regulates cardiomyocyte **differentiation***

3. The investigators use a heterologous system to show that Nkx (along with Tbx5 and GATA4) can activate Hey 2 transcription. The data shown shows relative changes in Hey expression compared to untransfected cells. It is critical to show on the same graph with the same scale the relative Hey2 expression levels in cardiomyocytes to ensure that the effect that is observed in the physiologically relevant range.

These experiments were designed to test the hypothesis that NKX2-5 is able to directly activate the enhancer elements identified by ChIP-seq. This data is the result of transient transfection experiments performed in HEK 293T cells. Importantly, the read-out of these assays is luciferase activity, not HEY2 expression level. Both the 5' and the 3' HEY2 enhancers are cloned into vectors containing a minimal TK promoter controlling luciferase and the data is reported relative to levels observed in cells transfected with the reporter and empty expression vectors. We make clear in the manuscript that this is a heterologous system that does not use hESC-derived cardiomyocytes. Thus, we are unable to directly address the reviewer's concern regarding HEY2 levels in cardiomyocytes in the manner requested, since the experiment does not measure HEY2 expression directly.

We are conscious of the caveats of using this approach and in particular the stoichiometry of transcription factors at these enhancers based on the use of expression vectors. As we note in the discussion "*It is likely that HEY2 regulation is multifactorial and complex*". Notably, enhancer elements function within a tightly regulated chromosomal architecture that is not recapitulated in transient transfection experiments. Nevertheless, these data support the notion that NKX2-5 functions at these enhancers *in vivo*. Furthermore, we show that both HEY2 transcript (Figs. 3a,b; 5 a,b,e; Supplementary Fig. 5c) and HEY2 protein levels (Supplementary Fig. 5e) are reduced in NKX2-5 knockout cardiomyocytes. In addition,

expression of HEY2::ER from the GAPDH locus is sufficient to rescue some aspects of the NKX2-5 null phenotype (Fig. 5f-h). In combination, our data support the hypothesis that *HEY2* is likely to be a direct target of NKX2-5 and an important transcriptional mediator of NKX2-5 function.

4. The reviewers should explicitly acknowledge in the discussion that the observed contractility defects may not be primary and may in fact be all secondary to conduction defect. Without the single cell analysis, it is simply not possible to conclude that there is a primary contractility defect.

Dissecting the electro-mechanical coupling in mutant cardiomyocytes is likely to be difficult in the context of the *NKX2-5* knockout due to the large number of differentially regulated genes. While the reviewer may be correct that the deficiency in contractile force is secondary to the abnormal electrophysiology demonstrated in *NKX2-5* null cultures our data do not allow us to explicitly conclude this is the case. In particular, the ventricular myogenic program is dysregulated (Fig. 3a,b; Fig. 4 b,e-g) and thus, cardiac sarcomeres in *NKX2-5* null cardiomyocytes are compromised (Fig. 2i, Supplementary Fig. 2g). Therefore, we cannot conclusively assign the myogenic force deficiency to impaired electrophysiology. For these reasons we present a carefully balanced discussion that avoids assigning a clear mechanistic explanation favouring an electrical rationale over compromised muscle integrity. As noted by the Reviewer, future mechanistic studies will require sophisticated single cell phenotyping to identify the key molecules driving the myogenic phenotype observed.

Reviewer #3 (Remarks to the Author):

The authors have addressed my concerns by indicating that the current cell culture model provides little (or no insight) into the mechanism by which haploinsufficiency of *NKX2-5* causes disease. The authors have adequately responded to other, noted, concerns.

We are pleased we could address the majority of Reviewer 3's concerns. Evidence to support the hypothesis that *NKX2-5* heterozygosity impairs cardiomyocyte differentiation or function is sparse. The vast majority of pathogenic *NKX2-5* mutations are proposed to act as dominant negative proteins, rather than resulting in haploinsufficiency. Thus, it is likely that differentiating human pluripotent stem cell models will offer mechanistic insights when *NKX2-5* mutations that underlie congenital heart defects are examined in this system.